# Effects of electric vehicle charging stations on the economic vitality of local businesses

Yunhan Zheng [1,2] ✉, David R. Keith[3], Shenhao Wang [4], Mi Diao [5] & Jinhua Zhao [6]

Electric vehicle charging stations (EVCS) are essential for promoting cleaner transportation by facilitating electric vehicle recharging. This study explores their broader economic impact on nearby businesses, analyzing data from over 4000 EVCS and 140,000 business establishments in California. Results show that installing one EVCS boosts annual spending at a nearby establishment by 1.4% ($1,478) in 2019 and 0.8% ($404) from January 2021 to June 2023. The effect is more pronounced when a point of interest (POI) is within 100 meters of an EVCS, with spending increasing by 2.7% in 2019 and 3.2% from January 2021 to June 2023 for that POI. Public EVCS tend to attract higher-income, exploratory visitors, and local residents. Moreover, they notably enhance businesses in underprivileged areas, defined as disadvantaged and/or low-income areas designated by both California and Justice40, indicating the importance of expanding EVCS in such communities. This study highlights EVCS as drivers of local economic growth and stresses the economic benefits of multi-host EVCS setups.

Electric vehicles (EVs), when integrated with low-carbon electricity production, offer a transformative solution to curbing greenhouse gas emissions, paving the way for a cleaner and more sustainable transportation future[1–3]. Alongside this transformation, the installation of public EV charging stations (EVCS) has played an important role in facilitating the transition to cleaner mobility options[1,4]. In recent years, there has been substantial growth in the development of EV charging infrastructure within the United States. A major milestone in this endeavor was the passage of the Infrastructure Investment and Jobs Act (IIJA) on November 15, 2021, allocating $7.5 billion towards the creation of a comprehensive charger network across the nation[5]. This landmark legislation underscores the government's commitment to addressing the climate crisis and advancing the clean energy sector. However, the importance of public EVCS extends beyond their primary function of providing a reliable charging infrastructure for electric vehicles.

Public EVCS have the potential to exert a broader influence on local communities, particularly on the economic vitality of surrounding businesses. As EV drivers park their vehicles to recharge, they often find themselves with spare time, creating an opportunity in activities such as shopping or dining in nearby establishments[6]. In addition, since people often have flexibility where they shop, having charging facilities available can make businesses more attractive to potential customers. This increased foot traffic can breathe new life into local businesses and may offer a substantial boost to their customer base and revenue[6,7]. Consequently, providing EVCS offers business owners the potential to diversify their income streams. Understanding the extent of this phenomenon is crucial for policymakers, EVCS providers, and business owners to harness the full potential of EV charging infrastructure and create sustainable and vibrant communities.

While there has been extensive research on the environmental benefits of EVCS, a noticeable research gap exists concerning the impact of these stations on local businesses and foot traffic. Recent studies conducted by EV charging companies have documented an

[1]Department of Civil and Environmental Engineering, Massachusetts Institute of Technology, Cambridge, MA, USA. [2]Singapore-MIT Alliance for Research and Technology Centre (SMART), Singapore, Singapore. [3]Melbourne Business School, The University of Melbourne, Carlton, VIC, Australia. [4]Department of Urban and Regional Planning, University of Florida, Gainesville, FL, USA. [5]College of Architecture and Urban Planning, Tongji University, Shanghai, China. [6]Department of Urban Studies and Planning, Massachusetts Institute of Technology, Cambridge, MA, USA. ✉e-mail: yunhan@mit.edu

increase in customer visits and spending at nearby businesses based on pilot experiments or surveys[8,9]. However, these studies are often limited in scale and lack causal analyses, making it challenging to generalize the results and differentiate the effects of EV charger installations on local businesses from the influence of other confounding factors, such as changes in economic, demographic, and business conditions. Another relevant area of research has explored the relationship between traditional transportation infrastructure and the consumption patterns of surrounding businesses. Studies focusing on new rail and subway stations[10,11], bus stops[12], and street configurations with increased accessibility[13] have consistently found positive impacts on retail sales. However, there has been a notable absence of research specifically addressing the influence of EVCS in this context.

In this work, we quantify the impact of installing EVCS on customer counts and spending in nearby businesses in California, a vanguard in both EV adoption and charging infrastructure deployment in the United States. Our analysis spans two distinct periods: 2019 and January 2021 to June 2023, aiming to mitigate the influence of the COVID-19 pandemic and anomalous EVCS counts during data integration in 2020 (see the Data section in Methods). Figure 1 shows the distribution of EVCS across California, with detailed views of Downtown San Francisco and Downtown Los Angeles during these periods. To mitigate endogeneity concerns, we employed a Difference-in-Differences (DID) methodology in conjunction with propensity score matching. We also examined variations in these effects based on different EVCS types, business types, and customer profiles. Additionally, by estimating the marginal benefit of proximity to an EVCS on local businesses, we quantified the cumulative impact of EVCS on customer expenditures within all nearby establishments.

## Results

To assess the impact of EVCS installations on local businesses, we implement an identification strategy that leverages spatiotemporal variations in customer counts and spending at points of interest (POIs) surrounding these stations, enabling the establishment of causal relationships. Our analysis incorporates three categories of POIs: (1) accommodation and food services; (2) retail trade; (3) arts, entertainment, and recreation. To account for variations in the timing of EVCS installations during our study periods, we have employed a DID methodology, controlling for POI-specific and county-by-month fixed effects. Our analytical process is distinctively structured for two study periods, namely the year 2019 and the period spanning January 2021 to June 2023. Within this framework, our treatment group encompasses POI locations featuring newly introduced EVCS within 500 m during the study period. In contrast, the control groups are selected from POIs devoid of proximate EVCS that were commissioned during the study period.

One potential limitation to the validity of this approach is that EVCS providers might strategically place EVCS in locations where the benefits of EVCS deployment are maximized, potentially impacting our results. To address this endogeneity concern, we employ propensity score matching, pairing treatment POIs with control POIs of the same category and similar characteristics (see spatial distributions of treated and control POIs in Downtown San Francisco and Downtown Los Angeles in Fig. 1c–f). Our formal analysis then focuses exclusively on the matched POIs, resulting in a smaller but more reliable sample for analysis. The summary statistics for the variables about treatment, control, and unmatched POIs are presented in Supplementary Tables S1 and S2.

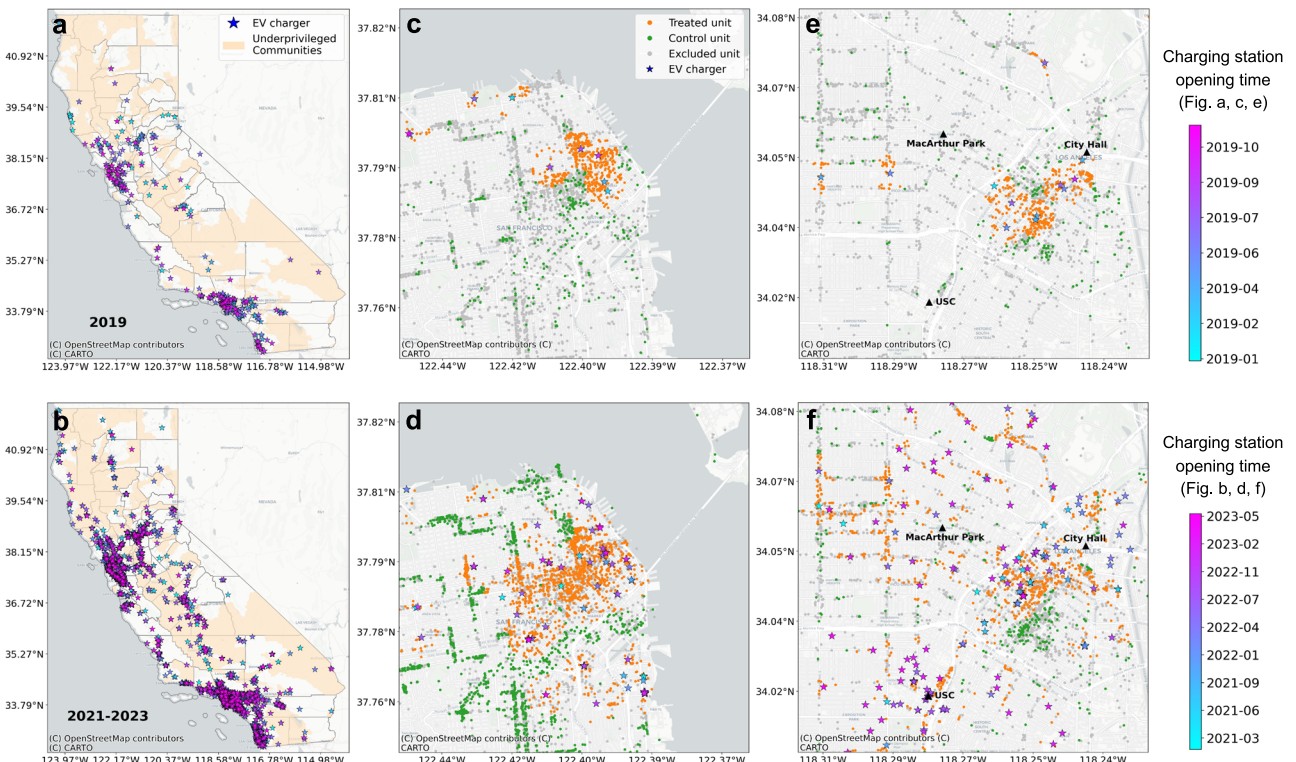

**Fig. 1 | Locations of EVCS in California and locations of treated and control points of interest (POIs) in Downtown San Francisco and Downtown Los Angeles. a** Locations of EVCS opened in 2019, **b** EVCS opened between February 2021 and June 2023. The areas shaded in pale orange represent underprivileged communities, which are defined as disadvantaged and/or low-income areas designated by both California and Justice40[14]. The locations of treated (orange) and control (green) POIs in Downtown San Francisco and Downtown Los Angeles are shown for different periods: Downtown San Francisco in 2019 (**c**), Downtown San Francisco from 2021 to 2023 (**d**), Downtown Los Angeles in 2019 (**e**), and Downtown Los Angeles from 2021 to 2023 (**f**). The base map layer is available from OpenStreetMap (openstreetmap.org/copyright).

**Table 1 | Impact of EV Charger Installations on Customer Count and Total Customer Spending**

| Dependent variables: | 2019 Sample | | | | 2021–2023 Sample | | | |
|---|---|---|---|---|---|---|---|---|
| | All Regions | | Underprivileged regions | | All Regions | | Underprivileged regions | |
| | Customer count | Spending | Customer count | Spending | Customer count | Spending | Customer count | Spending |
| | (1) | (2) | (3) | (4) | (5) | (6) | (7) | (8) |
| Treatment effect | 0.0021*** | 0.0025*** | 0.0017*** | 0.0029*** | 0.0014*** | 0.0016*** | 0.0008*** | 0.0009** |
| | (0.0004) | (0.0005) | (0.0006) | (0.0009) | (0.0002) | (0.0003) | (0.0003) | (0.0004) |
| | $p < 0.001$ | $p < 0.001$ | $p = 0.00883$ | $p = 0.00191$ | $p < 0.001$ | $p < 0.001$ | $p = 0.00784$ | $p = 0.0277$ |
| *Fixed-effects* | | | | | | | | |
| Individual POI | Yes | Yes | Yes | Yes | Yes | Yes | Yes | Yes |
| County-by-month | Yes | Yes | Yes | Yes | Yes | Yes | Yes | Yes |
| *Fit statistics* | | | | | | | | |
| Observations | 133,649 | 133,649 | 59,531 | 59,531 | 1,235,819 | 1,235,819 | 548,166 | 548,166 |
| R² | 0.96236 | 0.93579 | 0.96106 | 0.93387 | 0.89913 | 0.86249 | 0.89588 | 0.85679 |

*Note:* Clustered (at the POI level) standard-errors reported in parentheses, and *p*-values from two-sided *t*-tests are listed under standard errors. The dependent variables are the natural log of the number of customers and the natural log of total customer spending, respectively. ** $p < 0.05$; *** $p < 0.01$.

Table 1 shows the estimated impacts of installing a single EV charging port on the percentage increase in customer counts and spending at surrounding POIs. The analysis examines effects across all regions, with a particular focus on underprivileged regions. These regions are defined as disadvantaged and/or low-income communities according to designations by both California and Justice40[14]. Given the emphasis on equity in both federal and California government support for EVCS deployment[5,15], strengthened by the federal administration's Justice40 initiative[16], it is crucial to investigate the localized effects of EVCS installations on businesses in underprivileged communities.

Our findings consistently reveal significantly positive effects of newly installed EVCS on customer count and spending at surrounding POIs across all scenarios. Specifically, the introduction of an additional charging port resulted in a 0.21% increase in customer count and a 0.25% increase in spending in 2019. In the subsequent period spanning 2021 – 2023, although the effects are somewhat diminished, they remain statistically significant, yielding a 0.14% increase in customer count and a 0.16% increase in spending. The moderation in magnitude during this period might be attributed to factors such as constraints on customer buying power influenced by the COVID-19 pandemic, as well as variations in the utilization rate of public EV chargers over time.

It is noteworthy that when considering the average number of ports in a single EVCS, which was 5.4 in 2019 and 5.0 in 2021–2023, our findings indicate a more substantial impact. Specifically, the addition of a single EV charging station leads to a 1.4% increase in spending in 2019 and a 0.8% increase in spending in 2021–2023. While these effects may seem minor, they carry substantial importance given the context of low EV adoption rates (2.61% for Battery Electric Vehicles and 1.15% for Plug-in Hybrid Electric Vehicles in California by the end of 2022[17]) and the typically low utilization of EVCS (often averaging fewer than one session per port per day in the U.S.[18–20]).

Within underprivileged regions, the effects persist and maintain statistical significance. In 2019, an additional charging port led to a 0.17% rise in customer count and a 0.29% increase in spending. Between 2021 and 2023, the effects, although relatively smaller, remain notable, resulting in a 0.08% increase in customer count and a 0.09% increase in spending. We also conducted a sensitivity analysis, defining underprivileged communities based on designations from California and Justice40 separately. The estimated treatment effects remain significant and consistent in magnitude with the main specification (detailed in Supplementary Section 7).

## Spatial and temporal variations in treatment effects

To provide a more detailed understanding of our estimated treatment effects, we conduct a comprehensive analysis of how these effects vary across both spatial and temporal dimensions. We first categorize the distances between treatment POIs and their nearby EVCS into five distance bins, ranging from 0 to 500 m with 100 m increments. We investigated how the treatment effect differed across these distinct distance bins (refer to the model details in the Methods section).

Our analysis reveals that the magnitude of treatment effects varies considerably across different distance bins from EVCS, as depicted in Fig. 2a (for 2019) and 2b (for 2021–2023). As may be expected, the closest distance bin (0–100 m) exhibits the most substantial effect. Specifically, the introduction of an additional charging port within this proximity results in a noteworthy increase of ~0.5% (in 2019) and ~0.6% (in 2021–2023) in both customer counts and spending at local businesses. Conversely, as the distance from EVCS increases to the range of 400–500 m, the magnitude of the treatment effect diminishes, settling at ~0.2% for both customer counts and spending. The full results are reported in Supplementary Table S4. These findings align with urban planning literature which has established that convenience of access greatly influences consumer choices[13,21,22].

After estimating the marginal effect of adding one charging port, we find that the average marginal effect of adding one EV charging station on POIs within 100 m is quite substantial. This effect amounts to 2.7% in 2019 and 3.2% in 2021–2023 when multiplying the estimated marginal effect of adding one charging port with the average number of charging ports in a station.

To explore the temporal dynamics of the treatment effect, we implemented an event study strategy (detailed in the Methods section). Figure 2c (for 2019) and 2d (for 2021–2023) depict the marginal effects of an additional nearby EV charging port on the percentage change in spending. The results indicate that the marginal effects did not exhibit statistical significance before the installation of EVCS for both study periods, supporting the parallel trend assumption of the DID strategy. However, following the installation of EVCS, the marginal effect became significantly positive, validating the positive impact of EVCS installations on local business spending. It is noteworthy that in 2021–2023, the marginal effect declined in the later stage of the study period. This decline may be attributed to the saturation of the effect, particularly the decrease in utilization rate per EVCS port as the proliferation of EVCS may outgrow their demand. Alternatively, it could be influenced by behavioral factors, such as individuals' preference for exploring new destinations, diminishing the effectiveness of older EV

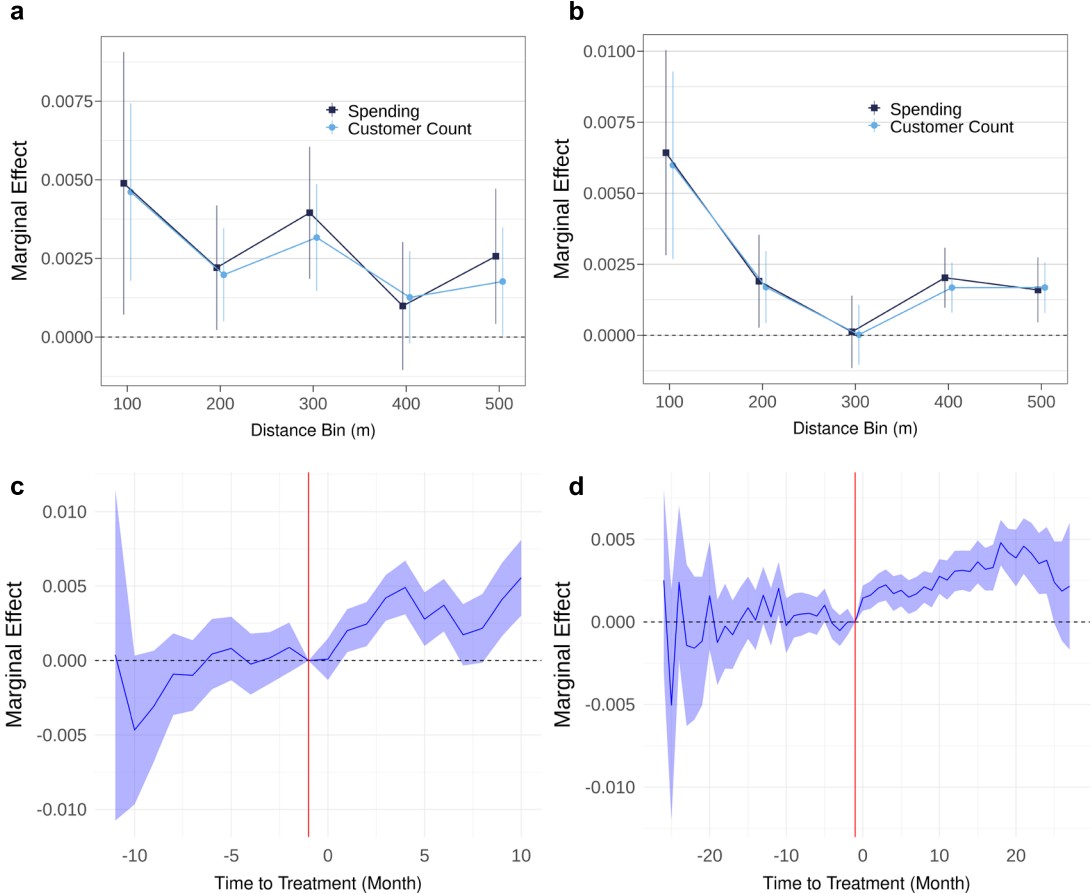

**Fig. 2 | Spatial and temporal variations in treatment effects.** Variations in treatment effects of EVCS installations on customer counts (light blue dots) and spending (dark blue squares) based on distance between the POIs and nearby EVCSs in 2019 (**a**) and 2021-2023 (**b**). Points represent point estimates of the treatment effects, and error bars denote the 95% confidence intervals. $n = 133,649$ observations (in 2019) and 1,235,819 observations (in 2021–2023). Time variation of the treatment effects on logged spending as analyzed through event studies in 2019 (**c**) and 2021-2023 (**d**), with the center of shaded error bands (blue lines) representing point estimates and error bands representing the 95% confidence intervals. The red vertical line represents the baseline period, which is one month before the treatment.

charging stations in attracting customers to surrounding businesses. Further research is necessary to pinpoint the precise reasons.

### Heterogeneity by EV charger types and POI types
Next, we delve into the treatment effect, stratifying by both EV charger type and POI category, with our findings presented in Table 2 (see the model details in the Methods section). Our examination reveals a noteworthy shift in treatment effects over time.

We observe that in the year 2019 treatment effects on spending were predominantly influenced by the presence of DC fast chargers. A plausible explanation for this phenomenon may be linked to the early stages of EV market development, characterized by a demographic skew towards higher-income families[23]. These individuals may have favored DC fast chargers for vehicle charging due to the higher utility offered compared to Level 2 chargers[24], and may tend to spend more at local businesses.

However, between 2021 and 2023, treatment effects on spending became predominantly associated with Level 2 chargers, rendering the influence of DC fast chargers statistically insignificant. This shift may be attributed to several factors. Notably, the years between 2021 and 2023 witnessed a substantial acceleration in the proliferation of Level 2 chargers compared to DC fast chargers, with the ratio of Level 2 chargers to DC fast chargers surging from 2.24 at the close of 2019 to 3.36 by the end of 2022 (see Supplementary Fig. S1). At the same time, as the EV market grew, there could be a broader and more diverse

EV user base, many of which may prefer Level 2 chargers. Another potential explanation is that as EVs matured over time, featuring longer driving ranges and increased charging station availability, the marginal need for DC fast charging diminished. Simultaneously, there has been an evolution in DC fast chargers' power output. In 2019, the majority of public DC fast charger ports operated at 50 kW or lower, whereas in 2021–2023, many DC fast charger stations have been upgraded to supply power ranging from 150 to 350 kW[20,25,26]. This increase in power output greatly reduces DC fast chargers' charging times, thereby diminishing the duration EV users spend waiting for their vehicles to charge. Conversely, the inherently slower charging rate of Level 2 chargers in contrast to DC fast chargers may have influenced individuals to allocate more time to activities such as shopping while their EVs are charging, which may have contributed to the pronounced effect of Level 2 EV charging stations on spending during this period.

Across various POIs, the impact of DC fast chargers in 2019 and Level 2 chargers in 2021–2023 on spending in local restaurants and grocery/clothing stores consistently demonstrates statistical significance. Interestingly, the EVCS installations had no significant effect on hotel spending in 2019, but from 2021 to 2023, the installation of Level 2 chargers significantly increased hotel revenue. This positive effect during 2021–2023 might be because hotels with EV chargers cater to EV drivers who need to recharge before continuing their journeys or returning home, thus attracting more guests and boosting

**Table 2 | Treatment effects on spending by POI types and EV charger types**

| Dependent variable: | Spending | | | | |
| --- | --- | --- | --- | --- | --- |
| | All | Restaurant | Grocery/clothing store | Hotel | Gasoline stations with convenience stores |
| | (1) | (2) | (3) | (4) | (5) |
| *(A) 2019 Sample:* | | | | | |
| Treatment effect (Level 2 chargers) | 0.0013 | 0.0001 | 0.0004 | -0.0087 | 0.0214 |
| | (0.0012) | (0.0016) | (0.0020) | (0.0072) | (0.0225) |
| | $p = 0.244$ | $p = 0.934$ | $p = 0.858$ | $p = 0.225$ | $p = 0.344$ |
| Treatment effect (DC fast chargers) | 0.0028*** | 0.0021* | 0.0021* | 0.0066 | 0.0065 |
| | (0.0007) | (0.0011) | (0.0011) | (0.0059) | (0.0075) |
| | $p < 0.001$ | $p = 0.0506$ | $p = 0.0619$ | $p = 0.265$ | $p = 0.387$ |
| Observations | 133,649 | 58,768 | 25,361 | 3,557 | 3,193 |
| $R^2$ | 0.93579 | 0.92946 | 0.94808 | 0.88982 | 0.94721 |
| *(B) 2021-2023 Sample:* | | | | | |
| Treatment effect (Level 2 chargers) | 0.0019*** | 0.0021*** | 0.0019*** | 0.0053*** | 0.0101*** |
| | (0.0004) | (0.0005) | (0.0007) | (0.0018) | (0.0038) |
| | $p < 0.001$ | $p < 0.001$ | $p = 0.00986$ | $p = 0.00308$ | $p = 0.00784$ |
| Treatment effect (DC fast chargers) | -0.0003 | 0.0002 | -0.0017 | 0.0043 | -0.0001 |
| | (0.0005) | (0.0007) | (0.0011) | (0.0029) | (0.0025) |
| | $p = 0.534$ | $p = 0.73$ | $p = 0.126$ | $p = 0.139$ | $p = 0.955$ |
| Observations | 1,235,819 | 625,211 | 187,609 | 30,531 | 39,877 |
| $R^2$ | 0.86250 | 0.82192 | 0.88485 | 0.77329 | 0.88141 |

*Note:* Clustered (placekey) standard-errors reported in parentheses, and *p*-values from two-sided *t*-tests are listed under standard errors. We controlled for individual POI and county-by-month fixed effects. *$p < 0.1$; ***$p < 0.01$.

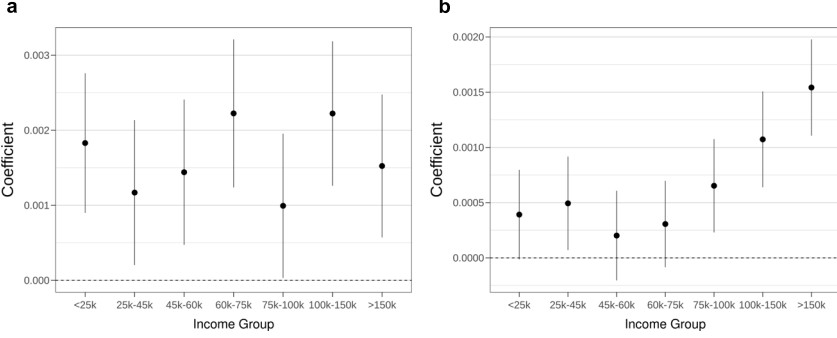

**Fig. 3 | Effects on populations from different income groups.** These figures illustrate the marginal effect of an additional EV charging port on customer counts among different income groups at surrounding businesses in 2019 (**a**) and 2021–2023 (**b**). Points represent point estimates of the effects, and error bars denote the 95% confidence intervals. $n = 133,649$ observations (in 2019) and 1,235,819 observations (in 2021–2023).

hotel spending[27]. Furthermore, from 2021 to 2023, the impact of Level 2 chargers on "gas stations with convenience stores" also became significant, suggesting that the presence of these chargers might have influenced consumer choices to visit convenience stores within gas stations during this period.

## Effects of EVCS on customer counts across different income groups

We also investigate the impact of EVCS installations on customer counts in nearby businesses among various income groups. Each customer falls into one of seven income groups: <$25 k, $25–45 k, $45–60 k, $60–75 k, $75–100 k, $100–150 k, and >$150 k, based on their annual income. These income groupings are determined using a proprietary model from Safegraph that classifies each customer based on their transactions and spending data (see the Methods section for details). As shown in Fig. 3a, the effect of EVCS on

customer counts was consistently positive across all income groups in 2019. Several factors could contribute to this trend. For example, prior studies have demonstrated that in California, as the proportion of multi-unit dwelling (MUD) housing units increases, so does the likelihood of access to public chargers[28]. Since low-income populations are more likely to reside in MUD housing units rather than single-family homes, they tend to have greater access to public chargers. Additionally, government incentive programs focusing on equity for both EVs and EVCS may have assisted low-income individuals in acquiring and charging EVs (see Supplementary Section 1.5). Furthermore, the rise in customer counts among low-income groups may also be partially attributed to the demand for charging services from low-income drivers affiliated with transportation network companies[29].

Conversely, in the period from 2021 to 2023, EVCS installations proved to be most effective in attracting customers among high-

**Table 3 | Effects on other variables**

| Dependent variables: | 2019 Sample | | | 2021–2023 Sample | | |
|---|---|---|---|---|---|---|
| | Distance from home | Median dwell time | Average customer income | Distance from home | Median dwell time | Average customer income |
| | (1) | (2) | (3) | (4) | (5) | (6) |
| Treatment effect | 0.0018[*] | 0.0007 | -0.0001 | -0.0073[***] | -0.0053[***] | 0.0003[***] |
| | (0.0010) | (0.0006) | (0.0001) | (0.0007) | (0.0005) | ($4.46 \times 10^{-5}$) |
| | $p = 0.0647$ | $p = 0.246$ | $p = 0.189$ | $p < 0.001$ | $p < 0.001$ | $p < 0.001$ |
| *Fixed-effects* | | | | | | |
| Individual POI | Yes | Yes | Yes | Yes | Yes | Yes |
| County-by-month | Yes | Yes | Yes | Yes | Yes | Yes |
| *Fit statistics* | | | | | | |
| Observations | 133,649 | 133,649 | 124,803 | 1,235,819 | 1,235,819 | 1,137,116 |
| $R^2$ | 0.90650 | 0.83894 | 0.72099 | 0.74681 | 0.66637 | 0.49003 |

*Note: Clustered (at the POI level) standard-errors reported in parentheses, and p-values from two-sided t-tests are listed under standard errors. [*]p < 0.1; [***]p < 0.01.*

income groups (Fig. 3b). During this time frame, the impact on customer counts from middle and low-income groups was relatively modest. This shift in pattern could be attributed to a combination of factors, including changes in the evolving EVCS network, shifting trends in EV adoption, evolving economic conditions, and shifts in consumer behavior, particularly in the aftermath of the COVID-19 pandemic. Further investigation is warranted to precisely identify the underlying reasons for this pattern shift. While previous studies have noted early EV adopters' tendency to be from higher-income populations, some research suggests broader diversification among EV buyers beyond high-income consumers (detailed in Supplementary Section 1.2). Therefore, it is possible that the considerably larger effect of EVCS on customer counts among high-income populations will subside as the population of EV adopters diversifies in the coming years.

## Effects on other variables

We conducted additional analyses to explore the impact of EVCS installations on several other variables, including customers' median distance from home, customers' median dwell time at surrounding businesses, and the average customer income at these businesses. The results are presented in Table 3.

Our analysis of 2019 data reveals insignificant or marginal significant effects on these variables. However, in the period from 2021 to 2023, EVCS installations had notable effects. They were associated with a decrease in customers' median distance from home and a reduction in customers' median dwell time. These findings suggest that the newly installed EVCS likely attracted customers residing in closer proximity, but these customers spent less time at the POIs compared to previous customers. This shift may indicate that the newly attracted customers are more inclined to be occasional or exploratory visitors rather than regular and loyal patrons. The presence of EVCS was associated with an increase in the average customer income at these businesses. This observation aligns with the common expectation, as the majority of newly attracted customers are likely to be EV owners, and it is well-documented that a considerable proportion of EV owners come from higher-income households[23].

We conduct additional analyses for different types of POIs regarding the effect on customers' median distance from home and median dwell time, the results of which are shown in Supplementary Tables S5 and S6. Notably, while EVCS installation is negatively associated with customers' median dwell time in restaurants, grocery/clothing stores, and hotels, it has a positive impact on customers' median dwell time in gas stations with convenience stores. One possible explanation is that because people spent more time charging

their EVs than refueling with gas, they tended to shop more at the convenience stores in those gas stations.

## Monetary impacts of EVCS on local businesses

To quantify the financial implications of EVCS openings on local businesses, we have converted our estimated effects into customer numbers and monetary values in US dollars (see Methods for details).

The estimated average treatment effect of adding one EVCS to a single surrounding POI is -17 additional customers and a spending increase of $1478 in 2019, and 5 additional customers and a spending increase of $404 per year between January 2021 and June 2023. These figures are calculated by multiplying the average pre-treatment customer count and spending across treated POIs by their estimated percentage change resulting from EVCS installation.

When we assess the average treatment effect of adding one EVCS across all surrounding POIs annually, the cumulative impact is substantial. In 2019, this cumulative impact amounts to ~$22,813, which is about 8.2% of the cost of infrastructure and installation for a 50kW DC fast charging station. In 2021–2023, as overall spending decreased after the COVID-19 pandemic and EVCS installations proliferated, this spending gain decreased to ~$3412 per year. Nevertheless, it still accounts for around 11.2% of the average infrastructure and installation cost of a Level 2 charging station (see Supplementary Note 8 for rough average EVCS cost estimates). When considering all EVCS installations in California during the study period, the total gain in customer spending due to all EVCS installed amounts to $6.7 million in 2019 and $19.5 million between January 2021 and June 2023. Detailed methods for calculating the monetary impacts are provided in the Methods section.

## Discussion

Operating charging infrastructure today presents considerable profitability challenges, and the societal impacts of EVCS are not yet fully understood. While government bodies and researchers have emphasized the potential benefits of EVCS in attracting customers and enhancing local consumption, a comprehensive, data-driven examination of the direct consequences of EVCS installations on local businesses has been notably absent. It is vital, therefore, to systematically discern and measure the influence of EVCS on customer visitation and expenditure at proximate POIs. This inquiry extends beyond mere convenience for existing customers who happen to be EV users; it seeks to determine whether the installation of EVCS attracts a broader customer base and contributes to increased spending at local establishments. Addressing this question holds broader importance—it not only elucidates the influence of EVCS as a burgeoning

transportation infrastructure on the economic vitality of local businesses but also imparts invaluable insights into the economic and operational models that underpin EVCS installations. Through this research, we bridge the gap between theoretical considerations and practical implications concerning the impact of EVCS on local economies, ultimately contributing to the advancement of sustainable transportation and urban planning.

Our results demonstrate that the effects of adding one additional charging port on customer count and spending in the vicinity of POIs consistently exhibit significance in both 2019 and 2021–2023. When translated into the average effect of EVCS, we observe that a POI with a newly-opened EVCS situated within 500 m experiences a 1.4% or $1478 per year increase in 2019 and a 0.8% or $404 per year increase in 2021–2023 in terms of customer spending. In particular, when the EVCS is positioned within 100 m, the marginal effect on spending exhibits even greater magnitude, standing at 2.7% in 2019 and rising to 3.2% in 2021–2023. While these monetary increments may appear modest at first glance, the cumulative impact of adding one EVCS becomes substantial when considering all proximate POIs: the introduction of a newly-opened EVCS leads to an average yearly total gain of $22,813 in customer spending in 2019 and $3412 in customer spending in 2021–2023. Taking into account all newly-opened EVCS in California during these periods, the benefits become pronounced, with the total gain in customer spending attributed to all EVCS installed amounting to $6.7 million in 2019 and $19.5 million between January 2021 and June 2023. Our findings provide compelling evidence that the positive impact of EVCS on local spending can partially offset the upfront infrastructure and installation costs that have historically posed a large hurdle to EVCS deployment. Moreover, the substantially greater benefit of EVCS on spending across all proximate POIs, in comparison to that of a single POI, highlights the advantages of multi-host EVCS installations, where several nearby hosts collaborate to site EVCS together, sharing both the costs and benefits.

Our analysis also demonstrates that the positive impacts of EVCS on businesses are not confined solely to high-income neighborhoods, where consumer purchasing power may already be relatively robust. Notably, EVCS can also significantly stimulate consumer spending in underprivileged areas. This discovery underscores the importance of policymakers actively endorsing the deployment of EVCS in marginalized areas. Not only do these EVCS installations offer immediate, first-order benefits like advancing sustainable mobility and fostering a cleaner environment, but they also hold the potential to serve as catalysts for enhancing the economic vitality of businesses operating in underprivileged areas.

While DC fast chargers played a more prominent role in inducing local spending in 2019, possibly due to the unique consumption patterns of early EV adopters, by 2021–2023, Level 2 chargers assumed a more substantial role as EVs and EVCS became more pervasive. This outcome underscores the potential influence of several factors associated with Level 2 chargers, including their affordability, broader accessibility within urban areas, and longer charging times. While federal IIJA funding and National Electric Vehicle Infrastructure (NEVI) programs primarily focus on building a national network of electric vehicle chargers along major highways to support long-distance trips[5], our results illuminate the growing importance of Level 2 chargers. These chargers provide convenient, low-cost charging options that can flexibly meet a wide range of daily needs, thereby holding the potential to invigorate local economies. The relatively lower cost of Level 2 chargers is also advantageous for EV charger providers, as they are often more financially feasible. Regarding DC fast chargers, no significant effect on local spending was found during 2021–2023. Future research should monitor the variability of their impact over time.

Across varying income strata, our study revealed a notable disparity: in the period of 2021–2023, the installation of EVCS resulted in a greater percentage increase in customer counts at local businesses among higher income brackets compared to lower income segments. This trend may be attributed to the prevailing tendency for EVs to be favored by individuals in higher income brackets[23,30,31]. However, it is plausible that this income-based variation in the effect of EVCS on customer counts will subside as the EV adopters diversify in the future.

This study contributes to the research on the business models of EV charger providers by highlighting the importance of integrating the benefits of EVCS on local businesses into the EVCS model. Furthermore, it aligns with previous literature that has explored the externalities of EVCS on various aspects such as EV adoption, air pollution, and housing prices—here, we demonstrate its positive externality on local businesses. Additionally, our research enriches the literature on the impacts of traditional transportation infrastructure on local businesses by illustrating the positive effects of EVCS as an emerging transportation infrastructure.

In practical terms, our research underscores the potential for policymakers to harness EVCS as a means to invigorate local economies, particularly in underprivileged regions. Furthermore, our findings suggest that EVCS providers can capitalize on the economic stimulus generated by these stations and develop a business model akin to the "gas station—convenience store" chain paradigm. Traditionally, many gas stations are affiliated with retail store chains, enabling owners to manage both fuel pumps and attached convenience stores, offering a diverse array of products beyond gasoline. These ancillary offerings often yield substantial profit margins[32]. By adopting a similar approach, EV chargers could strategically integrate with local businesses, thereby internalizing the positive impact of EVCS on the surrounding businesses.

This study also lays the foundation for future research endeavors. Firstly, while we made diligent efforts to mitigate endogeneity concerns by incorporating fixed effects, employing propensity score matching to identify the best matches with treated POIs, and conducting event analysis, the possibility remains that certain contemporaneous changes affecting the treated and control groups differently might have eluded detection. Subsequent investigations could explore alternative identification techniques, such as instrumental variables, to further scrutinize and validate our findings. Secondly, this study explores the net effect of EVCS on local spending, but further examination is needed to investigate the underlying causes of this effect on spending. Potential reasons could include EV drivers needing to charge and therefore visiting the POI, or visiting the POI regardless but staying for longer periods for their vehicles to charge. Finally, the second phase of our analysis covered the period from January 2021 to June 2023, a time when the impact of EVCS installations on local businesses may have been uniquely shaped by the COVID-19 pandemic. Hence, further research is crucial to assess the effects of EVCS installations on local businesses in the post-pandemic era.

## Methods
### Data
We collected data about EVCS from the United States Department of Energy's Alternative Fuels Data Center (AFDC)[33]. This dataset provides a repository of comprehensive information regarding EVCS across the United States, including various attributes such as geographical coordinates (longitude and latitude), installation dates, the count of ports for each type of EV charger (Level 1, Level 2, or DC fast chargers), and accessibility status (public or private). This dataset is currently the most comprehensive publicly available database for EVCS in the United States. However, it does have limitations, as some EVCS may not be updated in real-time within the dataset. For the scope of our analysis, we focus exclusively on public EVCS.

In this research, we exclusively consider EVCS that were opened either in 2019 or between February 2021 and June 2023. Several factors inform this choice: Firstly, data on customer visits and spending at POIs in California were only available starting from January 2019.

Therefore, conducting an analysis prior to 2019 is not feasible. Secondly, the disruptive impact of the COVID-19 outbreak in 2020 markedly affected the economy and businesses. Consequently, it is impractical to analyze the effect of EVCS on spending during that year. Lastly, there were anomalous surges in EV charging port counts due to the integration of data from EV charger providers into the data center in 2020 and January 2021, as illustrated in Fig. S3. These surges resulted in inaccuracies in the opening dates of EVCS during these periods. To mitigate the influence of this data management issue, we restrict our analysis to EVCS that were opened in 2019 or after January 2021 (i.e., from February 2021 to June 2023), allowing us to investigate how these EVCS impact local businesses while minimizing data inconsistencies.

While a considerable portion of our second study period coincides with the pandemic, we maintain that the pandemic has minimal impact on the validity of our identification strategy for the following reasons: firstly, vehicle movement in California has largely resumed since February 2021. In February 2021, vehicle miles traveled in California had returned to 95% of the levels observed in February 2019[34]. Secondly, the California stay-at-home order was lifted in January 2021, facilitating the resumption of gatherings and allowing outdoor operations for various businesses such as restaurants, hair salons, and nail salons[35]. Additionally, the vaccine rollout began in February 2021, with vaccination rates steadily increasing, particularly during the initial months. This rollout should have considerably alleviated COVID-related constraints on people's activities[36]. Furthermore, the inclusion of county-by-month fixed effects in our models should have accounted for any county-level time-varying factors that could influence spending in POIs, such as fluctuations in COVID-19 infections at the county level.

Information on POIs was sourced from Safegraph's Places data, offering location data and extensive attributions for each POI. This dataset includes details like geographical coordinates, POI categories, and the opening or closing dates of these establishments[37]. For our analysis, we exclusively considered POIs falling within the following three categories: (1) accommodation and food services; (2) retail trade; and (3) arts, entertainment, and recreation.

Monthly customer counts and spending data at these POIs were obtained from Safegraph's Spending data, derived from anonymized debit and credit card transactions[38]. Customer count represents the number of unique customers with at least one transaction at each POI in each month, while spending denotes the total amount spent at each POI across transactions during the same period. It's important to note that cash transactions are not captured by the dataset, which presents a limitation of the data. Additionally, the Spend dataset provides insights into customer income distribution. Specifically, each customer in the dataset is classified into an income class using a proprietary model based on his or her transactions and spending data[39]. We utilize this data to assess the impact of EVCS installation on customer spending across various income groups.

Customer metrics, including median distance from home and median dwell time at the POIs, were acquired from Advan's Pattern data and integrated with SafeGraph's Places data. This dataset originates from advanced multi-parameter models employed by Advan to analyze mobile phone location data[40].

Socio-demographic data at the census tract level, including population statistics, median household income, employed population, gender distribution, and ethnic composition, were derived from the American Community Survey (ACS). The locations of underprivileged communities (i.e. disadvantaged and/or low-income communities designated by both California and Justice40) come from California Energy Commission[41]. Data on EV sales at the county level was also obtained from the California Energy Commission[42]. Auto-oriented road miles per square mile and auto-oriented intersections per square mile at the block group level were sourced from EPA's Smart Location Database[43]. National Walkability Index score, which is a

measure of walkability score at the block group level, was obtained from the same database[43]. The building footprint data were obtained from OpenStreetMap[44].

For POI-related data, our analysis includes data from 2019 and the period between January 2021 and June 2023. It's worth noting that while the EVCS data for the latter period begins in February 2021, we have specifically included POI data in January 2021, treating it as a baseline period for the subsequent periods when EVCS openings potentially took place.

Data processing utilized Python version 3.7.4 and R version 3.6.3, while modeling was performed using R version 3.6.3.

## Identification strategies

The siting of EVCS is not random; it is influenced by regional factors like population, income levels, and road density, as noted in previous research[45–47]. Consequently, certain areas may exhibit a higher likelihood of hosting EVCS while simultaneously experiencing increased customer visits and spending. This raises concerns about self-selection bias and omitted variable bias when attempting to identify the causal effects of EVCS installations on local businesses[46].

To mitigate this issue, we employ a combination of three distinct approaches: First, we use propensity score matching (PSM), incorporating exact matching and nearest-neighbor matching strategies, to select control POIs closely resembling the treatment POIs. This method creates balanced comparison groups, enhancing the validity of our causal inferences. Second, we implement DID specifications that incorporate individual POI fixed effects and county-by-month fixed effects. This comprehensive approach not only accommodates the distinctive attributes of each POI but also captures dynamic factors influencing spending within the same county over time, such as economic fluctuations during the progression of the COVID-19 pandemic. Third, we employ event study analysis to assess preexisting differences in trends between the treatment and control groups. This analysis helps us examine and quantify any disparities in trends prior to the installation of EVCS. Detailed explanations of these strategies will be provided in subsequent sections.

## Propensity score matching (PSM)

In this analysis, the treated and control POIs were defined separately for the periods of 2019 and 2021–2023. For each of these two time periods, treated POIs were identified as those situated within a 500 m radius of EVCS that were opened during that specific period. Conversely, untreated POIs were designated as those located outside the 500 m radius of any EVCS that were opened during the same period. When estimating treatment effects for underprivileged communities, we add an additional requirement: both treated and untreated POIs should be located within underprivileged communities.

Prior to undertaking the formal DID analysis, we carefully select control POIs from the untreated POIs using PSM with a two-step approach: First, we initiate an exact match based on POI categories to ensure that each treated POI is paired with a control POI from the same category, thereby enhancing comparability. Subsequently, we employ nearest-neighbor PSM to match treated POIs with control POIs that share similar characteristics. These characteristics include: (1) Built environment factors: We consider population density, building density, auto-oriented road miles per square mile, auto-oriented intersections per square mile, and walkability index. These factors are chosen because EVCS tend to be installed in areas with dense populations and buildings; auto-oriented road and intersection density indicate car accessibility[47], and walkability reflects the ease of reaching a nearby POI from a charging station; (2) Socio-demographic variables: We incorporate median household income, percentage of employed population, gender distribution, and race composition. These variables are likely to influence EV ownership and charging demand[23,47]; (3) EV sales per 1000 people: This metric directly impacts current and

future charging demand, thereby influencing the likelihood of EVCS installation in a given location. (4) POI-level variables: The average customer count and spending when EVCS were not yet adopted during the corresponding study period are included as covariates. This is because EVCS installations may prefer areas with high foot traffic and consumer activity.

To explore the relationships between these covariates and the treatment, we employ logistic regression, regressing the treatment variable on these covariates. Our model demonstrates strong performance, with high accuracy scores (0.946 for the 2019 analysis and 0.796 for the 2021–2023 analysis). The results of this regression analysis are detailed in Supplementary Table S3.

Following the matching process, we assess the covariate balance between the treatment and control groups through graphical representation. Supplementary Fig. S3 (for 2019 data) and S4 (for 2021–2023 data) depict these graphs, illustrating the achievement of a balanced covariate distribution between the treated and untreated groups. Notably, the difference between the probabilities of the POIs being treated in the treatment and control groups approaches zero, affirming the effectiveness of our matching strategy in establishing comparable groups for subsequent analysis.

## Difference-in-differences (DID)

Following the selection of control POIs using PSM, we incorporate the matched control POIs alongside the treated POIs and proceed with the DID analysis. The model specification is defined as follows:

$$\ln(Y_{it}) = \alpha + \beta D_{it} \times PC_{it} + u_i + \phi_c \times \omega_t + \epsilon_{it} \tag{1}$$

In this equation, $\ln(Y_{it})$ represents the natural logarithm of the outcome variable, which can be either the number of customers or the spending amount at POI $i$ during time $t$. To avoid errors when taking the logarithm of zero, we add a value of one to $Y_{it}$ before computing the logarithm. $\alpha$ represents the intercept. $\beta$ represents the treatment effect, quantifying the marginal impact of adding a new charging port at an EVCS on the number of customers or the spending amount at the respective POI. $D_{it}$ is a binary treatment indicator that takes on the value of 1 if an EVCS was opened between the start of the study period (January 2019 or February 2021, depending on the analysis period) and time $t$ and is located within 500 m of POI $i$. $PC_{it}$ represents the total count of charging ports that became operational between the start of the study period and time $t$ and are within a 500 m radius of POI $i$. $u_i$ denotes the POI fixed effects. $\phi_c \times \omega_t$ represents the county-by-month fixed effects. $\epsilon_{it}$ represents the error term.

The treatment effect is represented by $\beta$, as it quantifies the marginal impact of adding a new charging port at an EVCS on the number of customers or the spending amount at the respective POI.

## Distance-varying treatment effect

In addition to exploring the average treatment effect, we explore the treatment effect's fluctuations concerning the distance between POI and their adjacent EVCSs. To numerically characterize this dynamic effect influenced by distance, we present the model formulation as follows:

$$\ln(Y_{it}) = \alpha + \sum_{d=1}^{5} \beta_d Dis_{idt} \times PC_{idt} + u_i + \phi_c \times \omega_t + \epsilon_{it} \tag{2}$$

where $Dis_{idt}$ is a binary dummy variable that takes the value of 1 if an EVCS was opened between the start of the study period (January 2019 or February 2021, depending on the analysis period) and time $t$ and is located within distance bin $d$ of POI $i$. We have utilized five distance bins spanning from 0 to 500 m, with 100 m increments. $PC_{idt}$ denotes the total count of charging ports that became operational between the start of the study period and time $t$ that fall within distance bin $d$ of POI

$i$. Therefore, $\beta_d$ represents the treatment effect specific to that particular distance bin $d$.

## Event study analysis

To evaluate the validity of the parallel trend assumption between POI equipped with nearby EVCSs and those without, and to explore the temporal changes in the treatment effect, we employ an event study analysis. The presence of a parallel trend implies that in the absence of treatment, the difference between the "treatment" and "control" groups is constant over time, thereby bolstering the credibility of our difference-in-differences analysis. The formulation of the event study model is presented as follows:

$$\ln(Y_{it}) = \alpha + \sum_{j=-m}^{0} \beta_j D_{i(t+j)} \times APC_i + \sum_{j=2}^{q} \gamma_j D_{i(t+j)} \times APC_i + u_i + \phi_c \times \omega_t + \epsilon_{it}$$

$$\tag{3}$$

where $D_{i(t+j)}$ indicates whether the treatment got switched on at time $t+j$ for POI $i$. $q$ and $m$ represent the number of leads and lags. $APC_i$ represents the monthly average number of ports during the treatment period for a treated POI. Specifically, it is calculated as $APC_i = \sum_{t=1}^{q} PC_{it}/q$, where $q$ is the number of treated months for POI $i$. $u_i$ and $\phi_c \times \omega_t$ represents the POI fixed effects and county-by-month fixed effects. This event study specification thus examines how the treatment effect per charging port varies depending on the timing of the treatment.

## Treatment effects by EV charger types and POI types

When estimating the treatment effects stratified by both EV charger type and POI types, we employ the following model specification to distinguish between the treatment effects of Level 2 chargers and DC fast chargers:

$$\ln(Y_{it}) = \alpha + \beta_1 D_{it} \times PC_{it}^{level2} + \beta_2 D_{it} \times PC_{it}^{dc} + u_i + \phi_c \times \omega_t + \epsilon_{it} \tag{4}$$

Here, $PC_{it}^{level2}$ and $PC_{it}^{dc}$ represent the number of charger ports for Level 2 and DC fast chargers, respectively. $\beta_1$ and $\beta_2$ thus capture the treatment effects associated with Level 2 chargers and DC fast chargers. When examining the effects for each POI type (including restaurants, grocery/clothing stores, hotels, and gas stations with convenience stores), we subset the sample, including only treated POIs from that specific POI type, along with their paired control POIs.

## Treatment effects on populations from different income groups

When estimating the treatment effects on populations from different income groups, we employ the following model specification:

$$\ln(Y_{it}^g) = \alpha + \beta^g D_{it} \times PC_{it} + u_i + \phi_c \times \omega_t + \epsilon_{it} \tag{5}$$

Here, $Y_{it}^g$ represents spending for income group $g$ at POI $i$ during time $t$, where $g$ can take on one of seven possibilities: >\$150 k, \$100 k–150 k, \$75–100 k, \$60–75 k, \$45–60 k, \$25–45 k, and <\$25 k. Data on $Y_{it}^g$ is sourced from SafeGraph's Spending data. The estimated $\beta^g$ values for each income group $g$ are visualized in Fig. 3.

## Effects on other variables

We also explore the treatment effects of adding one EV charger port on several other variables, including customers' median distance from home, customers' median dwell time, and average customer income. The data for customers' median distance from home and median dwell time are directly obtained from SafeGraph's Spending data. The average customer income is calculated based on customers' income groups, assuming that each customer within a group has the average income of that income bracket.

## Monetary impacts of EVCS

The treatment effect of adding one EVCS to a single surrounding POI per year, measured in US dollars ($M$), is calculated as follows: $M = \hat{\beta}_s * a * S * 12$. Here, $\hat{\beta}_s$ represents the estimated marginal effect of adding one charging port on the monthly spending of a single surrounding POI from Equation (1), $a$ is the average number of charging ports per charging station, which is calculated as the average number of ports at the same location in the AFDC dataset, and $S$ denotes the average pre-treatment monthly spending across treated POIs. Similarly, the impact of adding one EVCS on the customer count in a single surrounding POI per year ($F$) is calculated as $F = \hat{\beta}_c * a * C * 12$. Here, $\hat{\beta}_c$ indicates the estimated marginal effect of adding one charging port on monthly customer count of a single surrounding POI from Equation (1), $C$ denotes the average pre-treatment monthly customer count across treated POIs, and $a$ is the average number of charging ports per charging station.

The treatment effect of adding one EVCS on yearly spending in all surrounding POIs, denoted by $M_{all}$, is determined by: $M_{all} = M * p$, where $p$ represents the average number of POIs within 500 m of an EVCS during the study period. In 2019, this average was 15, while in 2021–2023, it was 8. This average is calculated by dividing the total number of treated POIs by the total number of EVCSs installed during the study period.

To calculate the total gains in spending due to all EVCSs installed during the study periods, the formula used is $\sum_i^N \sum_t^T \hat{\beta}_s * S_i * k_{it}$, where $S_i$ represents the pre-treatment spending for treated POI $i$, $k_{it}$ represents the number of charging port open at time $t$ for $i$, $N$ denotes the total number of treated POIs in the respective study period, and $T$ denotes the total number of months during the study period.

## Reporting summary

Further information on research design is available in the Nature Portfolio Reporting Summary linked to this article.

# Data availability

The POI-related data used in this study are not publicly available but can be requested from Dewey (https://www.deweydata.io/). Data access is granted through a subscription, which can be obtained at https://www.deweydata.io/subscribe. The EVCS data are available from the United States Department of Energy's Alternative Fuels Data Center (https://afdc.energy.gov/fuels/electricity_locations.html#/find/nearest?fuel=ELEC). All other data have been sourced from publicly available channels, and the specific sources for each variable are detailed in the Data section within the Methods.

# Code availability

The code used for conducting the analysis is accessible on GitHub at https://github.com/zhengyunhan/EVCS_economic_vitality[48].

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

## Acknowledgements

This research is supported by the National Research Foundation (NRF), Prime Minister's Office, Singapore under its Campus for Research Excellence and Technological Enterprise (CREATE) programme. The Mens, Manus, and Machina (M3S) is an interdisciplinary research group (IRG) of the Singapore MIT Alliance for Research and Technology (SMART) centre. M.D. acknowledges the partial support from the Natural Science Foundation of Shanghai (23ZR1465100) and the Fundamental Research Funds for the Central Universities of China.

## Author contributions

Y.Z. conceived and designed the research, conducted the analysis and wrote the paper. D.R.K. performed part of the analysis and edited the paper. D.R.K., S.W., M.D. and J.Z. supported the research and offered revision comments. J.Z. supervised the research. All authors discussed the results and contributed to the final manuscript.

## Competing interests

The authors declare no competing interests.
