## [Peer Review File · Nature Communications]

Effects of Electric Vehicle Charging Stations on the Economic Vitality of Local BusinessesREVIEWER COMMENTS

Reviewer #1 (Remarks to the Author):

This study is fascinating and most of the results are consistent with theory and prior research. The authors are to be commended for deploying novel data sources to answer empirical questions in robust ways. Some details in the results and methods raise further questions that should be addressed:

1) Table S3. Why is the R-squared for 2021-23 weaker compared to 2019? What are the implications for the effectiveness of the selection strategy for this period of time? Were other potential controls considered to address this but excluded?

2) Table 1. The proposed explanations for weakening magnitudes over time make sense. What happened to the EV penetration between these periods in California? If it grew consistently with EVCS growth, the increasing presence of charging stations wouldn't have had too much of a damper on the effect.

3) Page 12, lines 266-269. It is hard to imagine how EVCS would improve economic vitality, unless you could demonstrate that these stores are owned by residents in disadvantaged areas, and employee people in these areas. The wording should be more precise: EVCS can enhance the economic vitality of businesses operating in disadvantaged areas. That does not automatically mean the people living there will be better off.

4) If the authors are concerned about COVID-19 impacts, why include any of 2021 at all?

5) Page 14, lines 371-372 it says, "When estimating treatment effects for disadvantaged communities, untreated POIs remained unchanged..." Theoretically, the untreated POIs are also likely to be in disadvantaged communities as defined by Justice40 given the variables used in the PSM, but this implies that this is not required. This is a potential weakness of the disadvantaged communities analysis. Why not re-run the PSM to get untreated POIs that are also in disadvantaged communities? You are adding a criterion to your treatment without also applying it to a control in this case. It should be corrected or convincingly justified.

6) The maps appear to indicate some sort of spatial pattern--treatment areas are growing out of downtown cores. In LA, the EVCS appear to have grown along major arterials. Do you have a sense of why this is (see point 10 below), and have you thought through the implications for the identification strategy?

7) Page 12, Line 264, you might change "well-developed region" to "high income neighborhoods."

8) When/if EVCS become ubiquitous, will these results likely hold? Can the authors hypothesize as to why or why not?

9) The income analysis (Figure 3) is poorly explained. Sentence 175 sets us up to assume we are looking at the effects on income groups. That is not what is modeled. What is modelled is POIs ability to attract different income groups. This section needs to be re-written. Further, where are the actual model results, why are they not in the supplemental information?

10) The income results also raise other questions. It's confusing to see that EVCS openings would increase foot traffic among <25k income group as much as the <150k group. Were EVCS's typically installed in tandem other investments, say in active travel infrastructure? Or are they more likely to have been installed in new affordable housing complexes? Did California have EVs-for-low-income program at this time? More contextual detail that might help illuminate this finding is needed, particularly details about any government programs funding EVCS installation and the award criteria they may have used (which would impact what the authors call the "self-selection" of sites). There's also the question of ridehailing and taxis, which may also help explain the income results.

Reviewer #2 (Remarks to the Author):

This is an interesting topic and very timely and I think it can be an influential paper. Many stakeholders are discussing how to create sustainable business models for charging infrastructure, this paper can contribute to this discussion and highlights the benefits to businesses in installing EVSE. While many stakeholders discuss the potential benefits to local businesses, I have not seen a causal study on this topic yet. I have some questions and comments below, most are minor.

The literature review is short and some areas are lacking. One thing discuss is charging behaviour, it is important to frame to research so that readers understand how EVs are normally charged and when and which EV drivers may be the ones who visit public EVSE. Any research on charging station locations and research on business cases/models for EV charging would also be valuable.

It is also important to consider the types of people who buy EVs, especially relating to your finding about higher income patrons. As some point that effect may subside as EVs become adopted by other consumer groups, though it may be many years from now.

Methods

From my perspective (a transportation researcher) the methods are appropriate for making causal claims provided the right variables are selected for matching. Some clarification is needed on some aspects, see below.

It would be nice to understand more about the transaction data from safegraph. Does this mean the data does not include any cash or other types of transactions? Where does the income data come from? Is it whatever is self-reported to the bank by the customer? How is the foot count data compiled?

Please clarify which definition of disadvantaged community you are using (and in figure 1). Are you using both CalEnviroScreen and Justice40 definitions? It seems like maybe, but that isn't clear.

These also define disadvantaged communities very differently and it may not always make sense to group them if you have done so.

isn't clear what factors you are using to match POIs with similar POIs. Is the list in line 378 to 383 the variables you used? I also see a list in the appendix, but this list (in the covariance balance figure) is different from the list in the paper. Please clarify what is used in the PSM and why did you include/exclude these. There aren't any built environment factors listed, e.g. proximity to major roads or junctions, walkability of the area, density, etc. These could hypothetically also impact how EVSE impacts businesses and is an important omission from my perspective. I would like to see more discussion of this (variables included or excluded, and some justification of your decisions) given the importance of the selection of these to perform the matching. I am also unclear what is being matched, I presume is it a business, but not many factors relating to the business are included in the matching.

ADFC is not a comprehensive database, it may be the most comprehensive publicly available database but it is missing many charging locations (as you know, based on the large number of added chargers at certain times). Though I have no better recommendation, other than acknowledge and understanding that.

Results

I think some readers may perceive your results as a small increase in spending, however considering the very small number of EVs on the roads and often underutilized nature of EVSE I think it is substantial. You may want to discuss that in the study. Even in California only around 3% of all vehicles on the road are PEV, and among that less are BEVs (which use public charging, whereas PHEVs tend to not)

It is curious that you find no effect of DCFC in the 2021-2023 analysis. Why do you think this is? Do DCFC users not spend at local businesses during their visits? This would be concerning. The reduction in spending from 2019 to 2021-2023 is also concerning. Could this be due to the higher number of EVSE? As the number of EVs on the road increase wouldn't spending due to EVSE increase?

I also cannot understand why the finding for convenience stores and L2 charging makes sense. Is it because there are no other places to visit while charging? I can't imagine people choosing to go to a gas station store if there are other options nearby. Perhaps this occurs in some isolated areas?

In figure 3 the axis decreases in income moving toward the right-hand side. At least in my field of research this is an unusual choice and may confuse readers.

Please provide a reference for the cost of DCFC installation mentioned on line 215.

Most of the values you report are for 1 EVSE impact on all surrounding POIs, but that isn't always clear. Maybe in the abstract just mention that the values reported are per EVSE.

In your interpretation of the change in impact of DCFC from 2019 to 2021 you may want to discuss changes in DCFC and BEV charging speed. In 2019 most DCFC would be 50kw, whereas in 2021-2023 many would be 150-350kw.

Discussion

The discussion is brief and needs more development, including how this related to other research and implications for policymakers and industry. The paper would benefit from a conclusion that follows the discussion.

The lack of spending increase for DCFC charging is concerning to me. If these do not increase spending this could undermine ideas people have for sustainable business models for installing them. But the finding that L2 does increase spending is perhaps positive given businesses could conceivably afford these (whereas they may not be able to afford DCFC).

One thing the study cannot do is determine the cause of the increase in spending. Was it because EV drivers needed to charge so visited the POI, or were vesting the POI anyway and stayed for longer, or some combination of both.

It is also important to consider the types of people who buy EVs, especially relating to your finding about higher income patrons (EV buyers are higher income than the general population). At some point that effect may subside as EVs become adopted by other consumer groups, though it may be many years from now.

You may want to discuss the benefits of co locating charging with business and how this could be a business model for EV charging companies. I suppose this is very similar to the business model for gas stations, but EV drivers have more time to spare than gas station visitors. You may also want to mention how gas stations operate (selling gas close to cost and making profit from concessions), and how this could be viable for EVs.

Response Letter

The authors are grateful for the valuable feedback provided by the reviewers and the editor. In the revised manuscript, we have made significant enhancements, including the incorporation of built environment variables into the Propensity Score Matching (PSM) models, refinement of the PSM approach for disadvantaged communities, addition of a comprehensive literature review section, refinement and elaboration of model interpretations, as well as the augmentation of discussion sections and policy implications, among many other improvements. The modeling results have been thoroughly updated based on the revised models. To facilitate the review process, major changes have been highlighted in yellow within the manuscript. In this response letter, we offer item-to-item responses to reviewers' comments, with our responses in red and reviewers' comments in black (below).

Reviewer 1

This study is fascinating and most of the results are consistent with theory and prior research. The authors are to be commended for deploying novel data sources to answer empirical questions in robust ways. Some details in the results and methods raise further questions that should be addressed:

Reply: Thank you for your review and the positive assessment of our study's overall quality. Below, we have addressed your comments and made the necessary improvements to enhance our work:

1.1 — Table S3. Why is the R-squared for 2021-23 weaker compared to 2019? What are the implications for the effectiveness of the selection strategy for this period of time? Were other potential controls considered to address this but excluded?

Reply:

The decrease in R-squared is likely due to the decreased predictability of spending following the COVID outbreak, since the pandemic introduced some degree of economic volatility, changing consumer behavior and leading to greater uncertainty in forecasting spending patterns. However, given that the reduction in R-squared is not substantial, which is only around 7.5% (from 93.50% to 85.97%), we are confident that this does not diminish the effectiveness of our selection strategy for 2021-2023.

Regarding the control variables, our model has already controlled for POI fixed effects and county-by-month fixed effects, effectively addressing the unique characteristics of each POI and capturing the dynamic factors influencing spending within the same county. With these factors thoroughly considered, we are confident in the sufficiency of our model and find it unnecessary to include additional control variables.

1.2 — Table 1. The proposed explanations for weakening magnitudes over time make sense. What happened to the EV penetration between these periods in California? If it grew consistently with EVCS growth, the increasing presence of charging stations wouldn't have had too much of a damper on the effect.

Reply: Thank you for highlighting the potential impact of EV penetration on our findings. In order to delve into this aspect, we present Figure 1, illustrating the growth trajectory of EV sales from 2021 to

2023 alongside penetration of EV charging infrastructure for the same period. While there was notable growth in EV sales during this period alongside an increase in the number of EV charging ports, it's important to recognize that the diminishing effect on surrounding businesses may persist for several reasons:

- **Utilization Rate of Public Chargers:** The surge in EV sales doesn't necessarily translate to a proportional increase in the utilization rate of public charging stations. Many factors can influence demand for public charging, including the increasing driving range of new EVs, and driver preferences for home or workplace charging over public charging. Moreover, the spatial distribution of newly installed chargers may not align optimally with charging demand. Hence, the uncertainty surrounding these factors also affects the utilization rate of chargers.
- **Behavioral Dynamics:** From a behavioral standpoint, people's inclination to explore new destinations could contribute to the diminishing effect observed. While the installation of new EV chargers initially attracts customers to nearby establishments, over time (e.g. two years after installation), these patrons may begin to venture towards alternative locales. Consequently, the efficacy of older EV charging stations in driving customer to surrounding businesses may dwindle.

Therefore, the increase in EV sales may not necessarily counteract the diminishing effect of EV charging stations on surrounding businesses after approximately two years. We have incorporated the above discussion into the "Result - Spatial and temporal variations in treatment effects" section:

"This decline may be attributed to the saturation of the effect, particularly the decrease in utilization rate per EVCS port as the proliferation of EVCS may outgrow their demand. Alternatively, it could be influenced by behavioral factors, such as individuals' preference for exploring new destinations, diminishing the effectiveness of older EV charging stations in attracting customers to surrounding businesses. Further research is necessary to pinpoint the precise reasons."

Figure 1: Newly sold EV trends and the count of EV charging ports

1.3 — Page 12, lines 266-269. It is hard to imagine how EVCS would improve economic vitality, unless you could demonstrate that these stores are owned by residents in disadvantaged areas, and employee people in these areas. The wording should be more precise: EVCS can enhance the economic vitality of businesses operating in disadvantaged areas. That does not automatically mean the people living there will be better off.

Reply: Thank you for bringing this to our attention. We've incorporated your feedback and revised the statement as follows: *"...Not only do these EVCS installations offer immediate, first-order benefits like advancing sustainable mobility and fostering a cleaner environment; they also hold the potential to serve as catalysts for enhancing economic vitality of businesses operating in disadvantaged areas."*

1.4 — If the authors are concerned about COVID-19 impacts, why include any of 2021 at all?

Reply: Thank you for your inquiry regarding the inclusion of the year 2021. We opted to include data from 2021 because our evidence indicates a significant resumption of vehicle movement among Californians during this time period. Additionally, the easing of policy restrictions on mobility due to COVID-19, coupled with the rollout of vaccines in 2021, prompted us to consider this year in our study. Let us explain in details:

- First, vehicle movement has largely resumed since February 2021. In February 2021, vehicle-miles traveled in California had returned to 95% of the levels observed in February 2019 [9].
- Second, the California stay-at-home order was lifted in January 2021, facilitating the resumption of gatherings and allowing outdoor operations for various businesses such as restaurants, hair salons, and nail salons [6].
- Additionally, the vaccine rollout began in February 2021, with vaccination rates steadily increasing, particularly during the initial months. This rollout should have considerably alleviated COVID-related constraints on people's activities [21].

Furthermore, the inclusion of county-by-month fixed effects in our models should account for any county-level time-varying factors that could influence spending in POIs, such as fluctuations in COVID infections at the county level. With these considerations in mind, we believe it is reasonable to include data from 2021 in our analysis.

We have incorporated these considerations in the "Methods - Data" section. Additionally, we acknowledge the necessity for ongoing research in the post-pandemic era and have added the following statement to the Discussion section:

"Finally, the second phase of our analysis covered the period from January 2021 to June 2023, a time when the impact of EVCS installations on local businesses may have been uniquely shaped by the COVID-19 pandemic. Hence, further research is crucial to assess the effects of EVCS installations on local businesses in the post-pandemic era."

1.5 — Page 14, lines 371-372 it says, "When estimating treatment effects for disadvantaged communities, untreated POIs remained unchanged..." Theoretically, the untreated POIs are also likely to be in disadvantaged communities as defined by Justice40 given the variables used in the PSM, but this implies that this is not required. This is a potential weakness of the disadvantaged communities analysis. Why not re-run the PSM to get untreated POIs that are also in disadvantaged communities? You are adding a criterion to your treatment without also applying it to a control in this case. It should be corrected or convincingly justified.

Reply: Thank you for your insightful suggestion. Following your recommendation, we have rerun the Propensity Score Matching (PSM) analysis with untreated POIs also located within disadvantaged communities. Additionally, we have updated the Method section and reported the new results accordingly.

1.6 — The maps appear to indicate some sort of spatial pattern—treatment areas are growing out of downtown cores. In LA, the EVCS appear to have grown along major arterials. Do you have a sense of why this is (see point 10 below), and have you thought through the implications for the identification strategy?

Reply: Thank you for this observation. According to existing literature [11], some EVCS are situated along major arterials in order to serve the fuel demand of travelers on those roadways, indicating potentially high traffic volume and demand for charging services [12]. We acknowledge that the accessibility of charging facilities can influence the placement of EVCS. Therefore, we have incorporated auto-oriented road density and auto-oriented intersections density as two variables in the PSM. The matching results demonstrate a balanced covariate distribution between the treated and untreated groups for all variables, validating the efficacy of our matching strategy in establishing comparable groups for identifying the subsequent effects of EVCS.

1.7 — Page 12, Line 264, you might change “well-developed region” to “high income neighborhoods.”

Reply: We have revised the wording to “high-income neighborhoods” as suggested.

1.8 — When/if EVCS become ubiquitous, will these results likely hold? Can the authors hypothesize as to why or why not?

Reply: Our study has identified the significant impact of EVCS on local businesses, demonstrating their ability to attract more customers and stimulate higher spending in the surrounding area. Consequently, it is probable that this positive effect on businesses will persist even as EVCS become more widespread.

Nevertheless, given the current low utilization rate of EV chargers and the anticipated proliferation of EVCS, there is a risk that the utilization rate may decline further, potentially leading to a diminishing impact on surrounding businesses. However, it is unlikely that this effect will diminish to zero.

Moreover, if there is a surge in EV sales and total charging demand, the average impact of individual EVCS on local spending could potentially increase.

In summary, the positive effect of EVCS on local businesses will likely persist, but the precise marginal effect of EVCS is uncertain, as it is contingent on the interplay between EVCS availability and demand.

1.9 — The income analysis (Figure 3) is poorly explained. Sentence 175 sets us up to assume we are looking at the effects on income groups. That is not what is modeled. What is modelled is POIs ability to attract different income groups. This section needs to be re-written. Further, where are the actual model results, why are they not in the supplemental information?

Reply: Thank you for highlighting the necessity to improve the writing of the income analysis section. We have re-written this section to make it clear that we are investigating the effects of EVCS on customer counts in nearby businesses across different income groups. We have also reported the full model results for this part in Supplementary Section 6.

1.10 — The income results also raise other questions. It’s confusing to see that EVCS openings would increase foot traffic among <25k income group as much as the <150k group. Were EVCS’s typically installed in tandem other investments, say in active travel infrastructure? Or are they

more likely to have been installed in new affordable housing complexes? Did California have EVs-for-low-income program at this time? More contextual detail that might help illuminate this finding is needed, particularly details about any government programs funding EVCS installation and the award criteria they may have used (which would impact what the authors call the "self-selection" of sites). There's also the question of ridehailing and taxis, which may also help explain the income results.

Reply: Based on the reviewer's inquiries, we have provided further contemplation on why EVCS openings increase customer counts among low-income groups as much as high income group in 2019. Here are the potential reasons:

(1) EVCS investments in multi-unit dwelling housing areas

While we find no evidence that EVCS installed in tandem active travel infrastructure, previous research showed that as the percentage of multi-unit dwelling (MUD) housing units increases, the probability of public charger access also increases [12]. This is not only because these places typically have a greater public charger need due to the lower access to home chargers, but also because the policies support the installations of chargers in those areas. The California Electric Vehicle Infrastructure Project (CALeVIP), a project-based grant program supporting the installation of public chargers funded by the California Energy Commission, provides additional funding for public chargers installed in MUD site on top of the base rebate [4]. Since low income people are more likely to live in MUD compared with high income population, the fact that EVCS are more likely to be installed in areas with high MUD concentration may attract low-income population living in those areas.

(2) Equity-focused government incentive programs

The reviewer also inquires if there are any EVs-for-low-income funding program at that time, and whether that may somehow contribute to significant effect of EVCS on local spending from low-income groups. Indeed, California has made significant efforts to promote equity in EV adoption through legislative measures such as Senate Bill 535 (California Global Warming Solutions Act of 2006: Greenhouse Gas Reduction Fund) and Assembly Bill 1550 (Greenhouse gases: investment plan: disadvantaged communities). These bills mandate that a minimum of 25% of the Greenhouse Gas Reduction Fund be allocated to state programs aimed at reducing greenhouse gas emissions in disadvantaged communities [12, 18]. Additionally, there are EV charging station incentive programs that offer higher or exclusive rebates for low-income and disadvantaged communities. In 2018 and 2019, 69% of funding provided by CALeVIP was invested in Disadvantaged Communities or Low-Income Communities [5]. This government funding directed towards EV and EVCS markets in disadvantaged regions may contribute to the observed increase in customer counts among low-income groups. We have included the introduction of these programs in Supplementary Section 1 "Literature review".

(3) EV charging demand by ride-hailing services

As noted by the reviewer, the impact of EVCS on customer counts among low-income groups could also stem from the demand for charging services by drivers associated with transportation network companies (TNCs). Prior studies indicate that from 2014 to 2018, TNC services accounted for 35% of the total energy demand at non-Tesla DC fast-charging stations for non-TNC electric vehicles, underscoring the significant charging needs of ride-hailing drivers [13]. Low-income EV ride-hailing drivers could potentially boost customer counts and spending at local businesses while charging their vehicles, thereby contributing to the notable impact of EVCS openings on customer counts among low-income groups.

We have incorporated discussions on these factors within Section “Results - Effects by populations from different income groups”.

Reviewer 2

This is an interesting topic and very timely and I think it can be an influential paper. Many stakeholders are discussing how to create sustainable business models for charging infrastructure, this paper can contribute to this discussion and highlights the benefits to businesses in installing EVSE. While many stakeholders discuss the potential benefits to local businesses, I have not seen a causal study on this topic yet. I have some questions and comments below, most are minor.

Reply: We appreciate your valuable comments and your recognition of the commendable aspects of our study. We have taken your feedback seriously and have made significant revisions to the article to address your concerns. Please find our detailed response below.

2.1 — The literature review is short and some areas are lacking. One thing discuss is charging behaviour, it is important to frame to research so that readers understand how EVs are normally charged and when and which EV drivers may be the ones who visit public EVSE. Any research on charging station locations and research on business cases/models for EV charging would also be valuable.

Reply: In response to the reviewer's recommendation, we have included an extensive literature review in Supplementary Section 1. This section covers various aspects of EV charging, including different charging modes, characteristics of EV buyers, EVCS usage patterns, business models of EV chargers, and government incentives for promoting EV adoption. It aims to provide the audience with a comprehensive understanding of typical EV charging methods, detailed usage patterns, and the economic aspects of EV chargers.

2.2 — It is also important to consider the types of people who buy EVs, especially relating to your finding about higher income patrons. As some point that effect may subside as EVs become adopted by other consumer groups, though it may be many years from now.

Reply: Thanks for your comment. We have added a section “S1.2 Characteristics of EV buyers” in the Supplementary “Literature review” Section, in which we detailed the characteristics of EV buyers according to the previous literature. We further enrich the interpretation in Section “Results - Effects of EVCS on customer counts across different income groups”:

“While previous studies have noted early EV adopters’ tendency to be higher-income populations, some research suggests broader diversification among PEV buyers beyond high-income consumers (detailed in Supplementary Section 1.2). Therefore, it’s possible that the considerably larger effect of EVCS on customer counts among high-income populations will subside as the EV adopters diversify in the future.”

Methods

From my perspective (a transportation researcher) the methods are appropriate for making causal claims provided the right variables are selected for matching. Some clarification is needed on some aspects, see below.

Reply: We appreciate your valuable comments and address your concerns as below.

2.3 — It would be nice to understand more about the transaction data from Safegraph. Does this mean the data does not include any cash or other types of transactions? Where does the income data come from? Is it whatever is self-reported to the bank by the customer? How is the foot count data compiled?

Reply: Thank you for the question regarding the transaction data. The dataset solely includes debit and credit card transactions, excluding cash transactions and other forms of payment. The customer income data is derived from a proprietary model analyzing customers' transaction and spending patterns; it is not directly reported by customers to the bank. Regarding customer count data, it represents the unique number of customers with at least one transaction at each POI per month.

For further elucidation on the transaction data, please refer to the "Method" section, where we have provided additional clarification as follows:

"Monthly customer counts and spending data at these POIs were obtained from Safegraph's Spending data, derived from anonymized debit and credit card transactions [20]. Customer count represents the number of unique customers with at least one transaction at each POI in each month, while spending denotes the total amount spent at each POI across transactions during the same period. It's important to note that cash transactions are not captured by the dataset, which presents a limitation of the data. Additionally, the Spend dataset provides insights into customer income distribution. Specifically, each customer in the dataset is classified into an income class using a proprietary model based on his or her transactions and spending data [19]."

2.4 — Please clarify which definition of disadvantaged community you are using (and in figure 1). Are you using both CalEnviroScreen and Justice40 definitions? It seems like maybe, but that isn't clear. These also define disadvantaged communities very differently and it may not always make sense to group them if you have done so.

Reply: In our main specification, disadvantaged regions are defined as disadvantaged and/or low-income communities designated by both California and Justice40. We've clarified this definition in both the "Result" section and Figure 1.

In addition, we also conducted a sensitivity analysis, defining disadvantaged communities based on designations from California and Justice40 separately. The estimated treatment effects remain significant and consistent in magnitude with the main specification. These results are detailed in Supplementary Section 7.

2.5 — It isn't clear what factors you are using to match POIs with similar POIs. Is the list in line 378 to 383 the variables you used? I also see a list in the appendix, but this list (in the covariance balance figure) is different from the list in the paper. Please clarify what is used in the PSM and why did you include/exclude these. There aren't any built environment factors listed, e.g. proximity to major roads or junctions, walkability of the area, density, etc. These could hypothetically also

impact how EVSE impacts businesses and is an important omission from my perspective. I would like to see more discussion of this (variables included or excluded, and some justification of your decisions) given the importance of the selection of these to perform the matching. I am also unclear what is being matched, I presume is it a business, but not many factors relating to the business are included in the matching.

Reply: Thank you for your valuable input regarding PSM. We have made the necessary revisions to the Method section to clarify the variables utilized for PSM and have addressed your concerns as follows:

(1) In response to your suggestion, we have incorporated built environment factors, specifically population density, building density, auto-oriented road miles per square mile, and auto-oriented intersections per square mile in PSM. We did not include walkability as a covariate since EVCS installation is primarily influenced by car accessibility rather than pedestrian factors.

(2) We have explicitly stated that the matching process pertains to POIs (i.e. businesses). In our approach to POI-related matching, we implement two key strategies: firstly, we've conducted an exact match based on POI categories to ensure that each treated POI is paired with a control POI from the same category, thereby enhancing comparability. Secondly, we integrate two POI-level variables into PSM: average customer count and spending prior to EVCS adoption during the study period. We include these variables considering the potential preference for EVCS installations in areas with high foot traffic and consumer activity. This represents our utmost effort in POI-related matching, and currently, there are no other available POI-related variables suitable for PSM.

Here is our description of the covariates utilized in PSM as outlined in the Method section: *“(1) Built environment factors: We consider population density, building density, auto-oriented road miles per square mile, and auto-oriented intersections per square mile. These factors are chosen because EVCS tend to be installed in areas with high population and building density. Additionally, auto-oriented road and intersection density reflect accessibility to cars [11]; (2) Socio-demographic variables: We incorporate median household income, percentage of employed population, gender distribution, and race composition. These variables are likely to influence EV ownership and charging demand [11, 14]; (3) EV sales per 1000 people: This metric directly impacts current and future charging demand, thereby influencing the likelihood of EVCS installation in a given location. (4) POI-level variables. The average customer count and spending when EVCS were not yet adopted during the corresponding study period are included as covariates. This is because EVCS installations may prefer areas with high foot traffic and consumer activity.”*

2.6 — AFDC is not a comprehensive database, it may be the most comprehensive publicly available database but it is missing many charging locations (as you know, based on the large number of added chargers at certain times). Though I have no better recommendation, other than acknowledge and understanding that.

Reply: In light of your comment, we have enhanced the description of the AFDC dataset and acknowledged its limitations. Specifically, we have integrated the following statement into the “Method - Data” section: *“This dataset is currently the most comprehensive publicly available database for EVCS in the United States. However, it does have limitations, as some EVCS may not be updated in real-time within the dataset.”*

Results

2.7 — I think some readers may perceive your results as a small increase in spending, however considering the very small number of EVs on the roads and often underutilized nature of EVSE I think it is substantial. You may want to discuss that in the study. Even in California only around 3% of all vehicles on the road are PEV, and among that less are BEVs (which use public charging, whereas PHEVs tend to not)

Reply: Thank you for your insightful perspective. In light of your comment, we've discussed it in the Result section: *"It is noteworthy that ... the addition of a single EV charging station leads to a 1.4% increase in spending in 2019 and a 0.7% increase in spending in 2021-2023. While these effects may seem minor, they hold considerable significance given the context of low EV adoption rates (2.61% for Battery Electric Vehicles and 1.15% for Plug-in Hybrid Electric Vehicles in California by the end of 2022 [3]) and the typically low utilization of EVCS (often averaging fewer than one session per port per day in the U.S. [1, 2, 15])."*

2.8 — It is curious that you find no effect of DCFC in the 2021-2023 analysis. Why do you think this is? Do DCFC users not spend at local businesses during their visits? This would be concerning.

Reply: After refining our PSM strategies, we discovered that DCFC significantly impacts spending at hotels. However, the effects of DCFC on other venues remain statistically insignificant. It's important to clarify that the lack of a significant treatment effect of DCFC from 2021 to 2023 for POIs other than hotels does not necessarily imply that DCFC users do not contribute to local businesses during their visits. Instead, it suggests that their spending may not differ substantially from that of an average customer at similar POIs without DCFC. Various factors, such as the proliferation of Level 2 chargers, a more diverse EV user base, and shorter charging times for DCFC due to improving EV technology and DCFC power output, could influence EV users' preferences away from DCFC. We provide a detailed discussion regarding potential reasons for these insignificant effects of DCFC in the "Results - Heterogeneity by EV charger types and POI types" section:

"However, between 2021 and 2023, treatment effects on spending became predominantly associated with Level 2 chargers. The influence of DC fast chargers was only significant on hotels, while their effects on other POI types became statistically insignificant. This shift may be attributed to several factors. Notably, the years between 2021 and 2023 witnessed a substantial acceleration in the proliferation of Level 2 chargers compared to DC fast chargers, with the ratio of Level 2 chargers to DC fast chargers surging from 2.24 at the close of 2019 to 3.36 by the end of 2022 (see Supplementary Figure S2). At the same time, as the EV market grew, there could be a broader and more diverse EV user base, many of which may prefer Level 2 chargers. Another potential explanation is that as EVs matured over time, featuring longer driving ranges and increased charging station availability, the marginal need for DC Fast Charging diminished. Simultaneously, there has been an evolution in DC fast chargers power output. In 2019, the majority of public DC fast charger ports operated at 50 kW or lower, whereas in 2021-2023, many DC fast chargers stations have been upgraded to supply power ranging from 150 to 350 kW [2, 7, 10]. This increase in power output significantly reduces DC fast chargers' charging times, thereby diminishing the duration EV users spend waiting for their vehicles to charge. Conversely, the inherently slower charging rate of Level 2 chargers in contrast to DC fast chargers may have influenced individuals to allocate more time to activities such as shopping while their EVs charged, which may have contributed to the pronounced effect of Level 2 EV charging stations on spending during this period. "

2.9 — The reduction in spending from 2019 to 2021-2023 is also concerning. Could this be due to the higher number of EVSE? As the number of EVs on the road increase wouldn't spending due to EVSE increase?

Reply: The decrease in the impacts of Electric Vehicle Charging Stations (EVCS) on spending from 2019 to 2021-2023 may be influenced by two factors: (1) changes in the utilization rate per charger, and (2) the extent to which public EV charging attracts people to spend in local businesses.

Regarding the first aspect, despite the growth in EV sales during this period, it may not necessarily translate to a proportional increase in the utilization rate of public charging stations. Several factors, such as the preference for home or workplace charging over public charging, as well as the distribution of newly installed EV chargers, can influence the utilization rate, leading to variations in utilization per charger.

As for the second factor, the diminished effect observed in 2021-2023 could be attributed to constraints on EV users' purchasing power resulting from the COVID-19 pandemic. In other words, following the outbreak of the pandemic, EV users may have adopted more conservative purchasing behaviors.

We have addressed these two factors in the result interpretation section: *"The moderation in magnitude during this period could be attributed to factors such as constraints on customer buying power influenced by the COVID-19 pandemic, as well as variations in the utilization rate of public EV chargers over time."*

2.10 — I also cannot understand why the finding for convenience stores and L2 charging makes sense. Is it because there are no other places to visit while charging? I can't imagine people choosing to go to a gas station store if there are other options nearby. Perhaps this occurs in some isolated areas?

Reply: Thank you for your question regarding the impact of L2 chargers on convenience stores. It's important to highlight that in the revised model, our results indicate that the effect of L2 chargers on gas stations with convenience stores is only marginally significant (p-value = 0.065). Therefore, we exercise caution in interpreting the findings and revise the interpretation accordingly:

"Although EVCS did not notably impact 'gas stations with convenience stores' in 2019, we have noticed a marginally significant effect of EVCS on this particular category during 2021-2023. This suggests a potential influence of level 2 chargers on consumer decisions to patronize convenience stores within gas stations in 2021-2023, but due to the borderline significance, additional research is necessary to validate and comprehend the underlying mechanisms driving this association."

In response to the reviewer's comment, it's plausible that the observed positive effect could be attributed to the fact that these stores are often situated in relatively isolated areas, providing few alternative options for consumers during charging. However, given the marginal significance, we refrain from drawing definitive conclusions based solely on these results. Further investigation is warranted to elucidate the complexities of this relationship.

2.11 — In figure 3 the axis decreases in income moving toward the right-hand side. At least in my field of research this is an unusual choice and may confuse readers.

Reply: Thanks for your valuable input. Based on your suggestion, we have revised the graph to make x-axis increasing in income moving toward the right-hand side.

2.12 — Please provide a reference for the cost of DCFC installation mentioned on line 215.

Reply: Thank you for bringing this to our attention. We have now included the reference regarding the cost of DCFC installation.

2.13 — Most of the values you report are for 1 EVSE impact on all surrounding POIs, but that isn't always clear. Maybe in the abstract just mention that the values reported are per EVSE.

Reply: Thank you for your comment. We have clarified the values reported in the abstract as follows: *“Our results reveal that the installation of one EVCS increases annual spending at an average nearby POI by 1.4% (\$1,527) in 2019 and by 0.7% (\$364) from January 2021 to June 2023. The most significant effect is observed when EVCS are within 100 meters of a POI, resulting in an average boost of 2.6% in 2019 and 2.7% from January 2021 to June 2023 for that POI.”*

To clarify further, the primary results we report focus on the impact of one EVCS on one surrounding POI. In the “Monetary impacts of EVCS on local businesses” section, we compute the impacts on all surrounding POIs.”

2.14 — In your interpretation of the change in impact of DCFC from 2019 to 2021 you may want to discuss changes in DCFC and BEV charging speed. In 2019 most DCFC would be 50kw, whereas in 2021-2023 many would be 150-350kw.

Reply: Thank you for providing this valuable input. We have included it in the manuscript when interpreting the change in impacts of DCFC:

“The influence of DC fast chargers was only significant on hotels, while their effects on other POI types became statistically insignificant. This shift may be attributed to several factors.....there has been an evolution in DC fast chargers power output. In 2019, the majority of public DC fast charger ports operated at 50 kW or lower, whereas in 2021-2023, many DC fast chargers stations have been upgraded to supply power ranging from 150 to 350 kW [2, 7, 10]. This increase in power output significantly reduces DC fast chargers' charging times, thereby diminishing the duration EV users spend waiting for their vehicles to charge.”

Discussion

2.15 — The discussion is brief and needs more development, including how this related to other research and implications for policymakers and industry. The paper would benefit from a conclusion that follows the discussion.

Reply: Thank you for your valuable suggestion regarding enhancing the discussion section. Upon reviewing the requirements outlined by Nature Communications, we've noted that the paper structure calls for a single integrated “Discussion” section, rather than separate “Discussion” and “Conclusion” sections. Typically, both the conclusion and discussion points are incorporated within this unified “Discussion” section.

We have enriched the discussion within the “Discussion” section, elaborating on how this study connects to existing research and its implications for policymakers and industry. Below is the newly added discussion:

“This study contributes to the research on the business models of EV charger providers by highlighting the importance of integrating the benefits of EVCS on local businesses into the EVCS model. Furthermore, it aligns with previous literature that has explored the externalities of EVCS on various

aspects such as EV adoption, air pollution, and housing prices - here, we demonstrate its positive externality on local businesses. Additionally, our research enriches the literature on the impacts of traditional transportation infrastructure on local businesses by illustrating the positive effects of EVCS as an emerging transportation infrastructure.

In practical terms, our research underscores the potential for policymakers to harness EVCS as a means to invigorate local economies, particularly in disadvantaged regions. Furthermore, our findings suggest that EVCS providers can capitalize on the economic stimulus generated by these stations and develop a business model akin to the “gas station - store” chain paradigm. Traditionally, many gas stations are affiliated with retail store chains, enabling owners to manage both fuel pumps and attached convenience stores, offering a diverse array of products beyond gasoline. These ancillary offerings often yield significant profit margins [17]. By adopting a similar approach, EV chargers could strategically integrate with local businesses, thereby internalizing the positive impact of EVCS on the surrounding businesses.”

2.16 — The lack of spending increase for DCFC charging is concerning to me. If these do not increase spending this could undermine ideas people have for sustainable business models for installing them. But the finding that L2 does increase spending is perhaps positive given businesses could conceivably afford these (whereas they may not be able to afford DCFC).

Reply: After refining our PSM strategies, we discovered that DCFC significantly impacts spending at hotels. However, the effects of DCFC on other venues remain statistically insignificant. We have addressed this finding in the Discussion section and emphasized the importance of ongoing research to track changes in the effect: *“With respect to DC fast chargers, while their influence on spending in hotels remained significant during 2021-2023, it was not significant for other venues during this period. Future research should persist in monitoring the variability of DC fast chargers’ impact over time.”*

Regarding L2 chargers, we agree with the reviewer’s assessment of their positive effect for EV charger providers and have included this in the Discussion section: *“The relatively lower cost of Level 2 chargers is also advantageous for EV charger providers, as they are often more financially feasible.”*

2.17 — One thing the study cannot do is determine the cause of the increase in spending. Was it because EV drivers needed to charge so visited the POI, or were visiting the POI anyway and stayed for longer, or some combination of both.

Reply: Thank you for your valuable input. We have incorporated this point as a future research direction in the Discussion section: *“This study also lays the foundation for future research endeavors.....Secondly, this study explores the net effect of EVCS on local spending, but further examination is needed to investigate the underlying causes of this effect on spending. Potential reasons could include EV drivers needing to charge and therefore visiting the POI, or visiting the POI regardless but staying for longer periods for their vehicles to charge.”*

2.18 — It is also important to consider the types of people who buy EVs, especially relating to your finding about higher income patrons (EV buyers are higher income than the general population). At some point that effect may subside as EVs become adopted by other consumer groups, though it may be many years from now.

Reply: Thank you for highlighting this crucial aspect. In response to your comment, we have included the following in the Discussion section: *“Across varying income strata, our study revealed a notable*

disparity: in the period of 2021-2023, the installation of EVCS resulted in a greater percentage increase in customer counts at local businesses among higher income brackets compared to lower income segments. This trend may be attributed to the prevailing tendency for EVs to be favored by individuals in higher income brackets [8, 14, 16]. However, it is plausible that this income-based variation in the effect of EVCS on customer counts will subside as the EV adopters diversify in the future."

2.19 — You may want to discuss the benefits of co locating charging with business and how this could be a business model for EV charging companies. I suppose this is very similar to the business model for gas stations, but EV drivers have more time to spare than gas station visitors. You may also want to mention how gas stations operate (selling gas close to cost and making profit from concessions), and how this could be viable for EVs.

Reply: Thank you for your suggestion. We have elaborated on this point in the Discussion section: *"In practical terms, our research underscores the potential for policymakers to harness EVCS as a means to invigorate local economies, particularly in disadvantaged regions. Furthermore, our findings suggest that EVCS providers can capitalize on the economic stimulus generated by these stations and develop a business model akin to the "gas station - store" chain paradigm. Traditionally, many gas stations are affiliated with retail store chains, enabling owners to manage both fuel pumps and attached convenience stores, offering a diverse array of products beyond gasoline. These ancillary offerings often yield significant profit margins [17]. By adopting a similar approach, EV chargers could strategically integrate with local businesses, thereby internalizing the positive impact of EVCS on the surrounding businesses."*

References

- [1] Ahmad Almaghrebi, Subhaditya Shom, Fares Al Juheshi, Kevin James, and Mahmoud Alahmad. Analysis of user charging behavior at public charging stations. In *2019 IEEE Transportation Electrification Conference and Expo (ITEC)*, pages 1–6. IEEE, 2019.
- [2] Brennan Borlaug, Fan Yang, Ewan Pritchard, Eric Wood, and Jeff Gonder. Public electric vehicle charging station utilization in the united states. *Transportation Research Part D: Transport and Environment*, 114:103564, 2023.
- [3] California Energy Commission. Light-duty vehicle population in california, 2024. <https://www.energy.ca.gov/data-reports/energy-almanac/zero-emission-vehicle-and-infrastructure-statistics/light-duty-vehicle>, last accessed on 2024-03-25.
- [4] Center for Sustainable Energy. Calevip 1.0 rebate, 2024. <https://calevip.org/find-project>, last accessed on 2024-03-21.
- [5] Center for Sustainable Energy. Rebate statistics dashboard, 2024. <https://calevip.org/rebate-statistics>, last accessed on 2024-03-21.
- [6] CNN. California lifts regional stay-at-home orders as icu crowding eases, 2021. <https://www.cnn.com/2021/01/25/us/california-coronavirus-stay-at-home-orders/index.html>, last accessed on 2024-03-21.
- [7] EVBox. The current state of the dc fast-charging market, 2023. <https://blog.evbox.com/state-of-dc-market>, last accessed on 2024-03-25.
- [8] Z Farkas, Hyeon-Shic Shin, and A Nickkar. Environmental attributes of electric vehicle ownership and commuting behavior in maryland: Public policy and equity considerations. *Mid-Atlantic Transportation Sustainability University Transportation Center. Retrieved April, 20:2019*, 2018.
- [9] FHWA. Traffic volume trends (february 2021), 2021. https://www.fhwa.dot.gov/policyinformation/travel_monitoring/21febtvt/21febtvt.pdf, last accessed on 2024-03-21.

- [10] Forbes. The future of ev charging may be at 50kw, not the ‘gasoline thinking’ of 250kw, 2020. <https://www.forbes.com/sites/bradtempleton/2020/07/09/the-future-of-ev-charging-may-be-a-t-50kw-not-the-gasoline-thinking-of-250kw/?sh=75b79d037535>, last accessed on 2024-03-25.
- [11] Sylvia Y He, Yong-Hong Kuo, and Dan Wu. Incorporating institutional and spatial factors in the selection of the optimal locations of public electric vehicle charging facilities: A case study of beijing, china. *Transportation Research Part C: Emerging Technologies*, 67:131–148, 2016.
- [12] Chih-Wei Hsu and Kevin Fingerma. Public electric vehicle charger access disparities across race and income in california. *Transport Policy*, 100:59–67, 2021.
- [13] Alan Jenn. Emissions benefits of electric vehicles in uber and lyft ride-hailing services. *Nature Energy*, 5(7):520–525, 2020.
- [14] Jae Hyun Lee, Scott J Hardman, and Gil Tal. Who is buying electric vehicles in california? characterising early adopter heterogeneity and forecasting market diffusion. *Energy Research & Social Science*, 55:218–226, 2019.
- [15] Patrick Morrissey, Peter Weldon, and Margaret O’Mahony. Future standard and fast charging infrastructure planning: An analysis of electric vehicle charging behaviour. *Energy policy*, 89:257–270, 2016.
- [16] Erich Muehlegger and David Rapson. Understanding the distributional impacts of vehicle policy: who buys new and used alternative vehicles? 2018.
- [17] NACS. Consumer behavior at the pump, 2019. <https://www.convenience.org/topics/fuels/documents/how-consumers-react-to-gas-prices.pdf>, last accessed on 2024-03-26.
- [18] OEHHA. Sb 535 disadvantaged communities, 2024. [https://oehha.ca.gov/calenviroscreen/sb535#:~:text=Senate%20Bill%20535%20\(De%20Le%C3%B3n,projects%20located%20within%20those%20communities](https://oehha.ca.gov/calenviroscreen/sb535#:~:text=Senate%20Bill%20535%20(De%20Le%C3%B3n,projects%20located%20within%20those%20communities), last accessed on 2024-03-21.
- [19] SafeGraph. Safegraph data: Customer information, 2023. <https://docs.safegraph.com/docs/spend#section-customer-information>.
- [20] SafeGraph. Safegraph data: Spend, 2023. <https://docs.safegraph.com/docs/spend>.
- [21] The Centers for Disease Control and Prevention. Covid-19 vaccinations in the united states, 2022. <https://data.cdc.gov/Vaccinations/COVID-19-Vaccinations-in-the-United-States-Jurisdiction/unsk-b7fc>, last accessed on 2022-09-12.

REVIEWER COMMENTS

Reviewer #1 (Remarks to the Author):

My concerns have been satisfactorily addressed.

Reviewer #2 (Remarks to the Author):

The revisions have mostly improved the paper, some minor comments below.

This comment doesn't make sense: "We did not include walkability as a covariate since EVCS installation is primarily influenced by car accessibility rather than pedestrian factors" You could include walkability as a measure of the ease of getting from a charger to a POI, not a measure of the ease of getting to a charger. Drivers can only visit places they can walk to if the route is walkable from the charger.

Can you give some context for customer counts, how many customers is a 0.13% or 0.27% increase?

If the EVCS increases spending at 1 POI by \$1500 to \$300, what does this mean for total POI spending increases due to the EVCS? It looks like there are an average of 26 POIs per EVCS.

Figure 1 isn't very useful, its hard to see much detail. I would at least have zoomed in panels for LA and the Bay Area.

The effect of DCFC on hotel spending in particular is hard to make sense of. Have you thought about an explanation of this finding?

There are grammatical errors and typos in the paper that need addressing. There are too many to individually list, please either proof read or use some language tools or AI to help.

It still isn't clear what your definition of a disadvantaged region is. DACs are not necessarily low-income communities, and California has a different definition of low income communities. It looks like you are looking at DAC as per CalEnviroScreen, DAC as per justice40, and low income as per California Department of Housing. The latter are not necessarily DACs, so you are looking at DAC and low income communities, or priority populations.

The discussion is long, the paper would benefit from a short conclusion with key takeaways from the paper.

Supplemental information

Some of the data in S1 and S2 is hard to follow. The number of customers is 130 and spending is \$9,530. Are these values per day, month, year? Same for EV sales, as these total sales or per year?

The included reference of estimated charging costs are from a 2015 report, which is far to old for a data source for EV charging costs. Please update this with newer data.

Response Letter

Reviewer 1

My concerns have been satisfactorily addressed.

Reply: Thank you for your comments! We appreciate the contributions from you, which have helped us to improve the manuscript significantly.

Reviewer 2

The revisions have mostly improved the paper, some minor comments below.

Reply: Thank you for acknowledging the revisions we made based on your earlier feedback and for providing additional constructive comments. We have addressed your comments as follows:

2.1 — This comment doesn't make sense: "We did not include walkability as a covariate since EVCS installation is primarily influenced by car accessibility rather than pedestrian factors" You could include walkability as a measure of the ease of getting from a charger to a POI, not a measure of the ease of getting to a charger. Drivers can only visit places they can walk to if the route is walkable from the charger.

Reply: Thank you for highlighting that point. We acknowledge that walkability can indicate the ease of accessing Points of Interest (POIs) from a charger, and we have now incorporated it into our Propensity Score Matching (PSM) models. The walkability indicator we used is the National Walkability Index, a metric provided by the US Environmental Protection Agency that assesses walkability at the block group level. The PSM analysis reveals a positive association between higher walkability and a greater likelihood of electric vehicle charging stations (EVCS) installation.

Following this revised PSM approach, we have updated all modeling results. The estimated effects remain largely consistent after the model refinement. For example, regarding our main results, the estimated effect of adding an extra charging port on spending in 2019 changed slightly from 0.29% (previous version) to 0.25% (revised version), while in the period from 2021 to 2023, the effect shifted slightly from 0.17% to 0.16%.

2.2 — Can you give some context for customer counts, how many customers is a 0.13% or 0.27% increase?

Reply: Thank you for this important question. We have added the implication of EVCS opening on customer counts of surrounding businesses in the "Monetary impacts of EVCS on local businesses" section:

"The estimated average treatment effect of adding one EVCS to a single surrounding POI is approximately 17 additional customers and a spending increase of \$1,478 in 2019, and 5 additional customers and a spending increase of \$404 per year between January 2021 and June 2023. These figures are calculated by multiplying the average pre-treatment customer count and spending across treated POIs

by their estimated percentage change resulting from EVCS installation.”. We have also detailed the formula used to calculate these effects in the “Methods” section.

2.3 — If the EVCS increases spending at 1 POI by \$1500 to \$300, what does this mean for total POI spending increases due to the EVCS? It looks like there are an average of 26 POIs per EVCS.

Reply: This means that the total POI spending increase due to the installation of one EVCS is \$22,813 in 2019 and \$3,412 in 2021-2023, which is calculated based on the fact that there is an average of approximately 15 POIs surrounding one EVCS in 2019 and 8 POIs in 2021-2023. We’ve discussed these estimates in the “Monetary impacts of EVCS on local businesses” section:

*“When we assess the average treatment effect of adding one EVCS across all surrounding POIs annually, the cumulative impact is substantial. **In 2019, this cumulative impact amounts to approximately \$22,813, which is about 8.2% of the cost of infrastructure and installation for a 50kW DC fast charging station. In 2021-2023, as overall spending decreased after the COVID-19 pandemic and EVCS installations proliferated, this spending gain decreased to approximately \$3,412. Nevertheless, it still accounts for around 11.2% of the average infrastructure and installation cost of a Level 2 charging station (see Supplementary Note 8 for rough average EVCS cost estimates).**”*

2.4 — Figure 1 isn’t very useful, its hard to see much detail. I would at least have zoomed in panels for LA and the Bay Area.

Reply: Thank you for your feedback. We’ve updated Figure 1 to also include detailed zoom-in panels for Downtown San Francisco and Downtown LA, showing the spatial distribution of EVCS, treated, and control POIs. We believe this improvement will provide a clearer view and make it easier for the audience to grasp the data.

2.5 — The effect of DCFC on hotel spending in particular is hard to make sense of. Have you thought about an explanation of this finding?

Reply: Thank you for pointing this out. Our updated modeling results indicate that the effect of DCFC on hotel spending is not significant in either 2019 or 2021-2023. However, we found a significant impact from Level 2 chargers on hotel spending during the 2021-2023 period. Here’s our interpretation of this result:

“The EVCS installations had no significant effect on hotel spending in 2019, but from 2021 to 2023, the installation of Level 2 chargers significantly increased hotel revenue. This positive effect during 2021-2023 might be because hotels with EV chargers cater to EV drivers who need to recharge before continuing their journeys or returning home, thus attracting more guests and boosting hotel spending [4].”

2.6 — There are grammatical errors and typos in the paper that need addressing. There are too many to individually list, please either proof read or use some language tools or AI to help.

Reply: Thank you for bringing this to our attention. We have thoroughly proofread the manuscript and ensure that these issues have been addressed.

2.7 — It still isn’t clear what your definition of a disadvantaged region is. DACs are not necessarily low-income communities, and California has a different definition of low income communities. It

looks like you are looking at DAC as per CalEnviroScreen, DAC as per justice40, and low income as per California Department of Housing. The latter are not necessarily DACs, so you are looking at DAC and low income communities, or priority populations.

Reply: Thank you for pointing out the ambiguity. Indeed, we recognize that Disadvantaged Communities (DACs) are not necessarily low-income communities, as different agencies use varying criteria for these classifications. To address this, we now use the term “underprivileged regions” to encompass both DACs and low-income communities. This broader term reflects communities that are either California-designated DACs, Justice40-designated DACs, or California-designated low-income communities. The exact definition is outlined in the “Results” section as follows:

“...The analysis examines effects across all regions, with a particular focus on underprivileged regions. These regions are defined as disadvantaged and/or low-income communities according to designations by both California and Justice40 [1].”

Additionally, we direct the audience to the California Energy Commission website [1] for detailed explanations of what constitutes disadvantaged or low-income communities designations by both California and Justice40 (i.e., our “underprivileged regions”).

2.8 — The discussion is long, the paper would benefit from a short conclusion with key takeaways from the paper.

Reply: Thank you for this suggestion. We have shortened the discussion section to focus on providing the key findings and takeaways from our study. The remaining discussions comprise a concise summary of the results, crucial insights derived from the paper, and essential discussions that were incorporated during the previous review round, following the recommendations from the reviewers.

In addition, we have verified that the total word count of the main text (including Introduction, Results, and Discussion) is 4,321, which is within the 5,000-word limit required by Nature Communications.

1 Supplemental information

2.9 — Some of the data in S1 and S2 is hard to follow. The number of customers is 130 and spending is \$9,530. Are these values per day, month, year? Same for EV sales, as these total sales or per year?

Reply: Thank you for pointing out the ambiguity. Regarding customer counts and spending, the values you mentioned (130 customers and \$9,530 spending) represent average monthly figures per POI. Similarly, for EV sales, these numbers represent average monthly EV sales. We have updated Tables S1 and S2 to clarify the units of measure.

2.10 — The included reference of estimated charging costs are from a 2015 report, which is far too old for a data source for EV charging costs. Please update this with newer data.

Reply: Thank you for pointing this out. We have updated the reference with newer data, specifically a 2019 report by the International Council on Clean Transportation (ICCT). We have also cross-checked the cost estimates from this report with other sources, including a 2019 study by the Rocky Mountain

Institute [2] and a 2015 report by the Department of Energy [3], and found general consistency in the range of estimates.

For a detailed explanation of the sources used for our cost estimates, please refer to Supplementary Note 8.

References

- [1] California Energy Commission. California and justice40 disadvantaged or low-income communities, 2024. https://cecgis-caenergy.opendata.arcgis.com/datasets/ec0adaef7db349dfa584ee33ea4c3f1f_0/explore, last accessed on 2024-03-21.
- [2] Chris Nelder and Emily Rogers. Reducing ev charging infrastructure costs. *Rocky Mountain Institute*, 17, 2019.
- [3] U.S. Department of Energy. Costs associated with non-residential electric vehicle supply equipment: Factors to consider in the implementation of electric vehicle charging stations. Technical report, 2015.
- [4] Lixian Qian and Cheng Zhang. Complementary or congruent? the effect of hosting tesla charging stations on hotels' revenue. *Journal of Travel Research*, 62(3):663–684, 2023.

REVIEWERS' COMMENTS

Reviewer #2 (Remarks to the Author):

The authors have addressed most of the comments. I think they misinterpreted this comment "The discussion is long, the paper would benefit from a short conclusion with key takeaways from the paper." I was suggesting the paper could benefit from a short conclusion section, rather than shortening the discussion. That suggestion (adding a conclusion) is still relevant.

Response Letter

Reviewer 2

The authors have addressed most of the comments. I think they misinterpreted this comment "The discussion is long, the paper would benefit from a short conclusion with key takeaways from the paper." I was suggesting the paper could benefit from a short conclusion section, rather than shortening the discussion. That suggestion (adding a conclusion) is still relevant.

Reply: Thank you for clarifying your comment. Nature Communications has specific requirements for the structure of papers (see <https://www.nature.com/ncomms/submit/article>). According to their guidelines, an article can include a Discussion section but should not include a separate Conclusion section. Typically, both conclusion and discussion points are incorporated within this unified "Discussion" section.

This is why we have included only a Discussion section, where the short conclusion has already been incorporated.